# Closed-loop two-photon functional imaging in a freely moving animal

Paul McNulty[1,5], Rui Wu [1,5], Akihiro Yamaguchi[1], Ellie S. Heckscher [2], Andrew Haas[1], Amajindi Nwankpa[1], Mirna Mihovilovic Skanata [1] ✉ & and Marc Gershow [1,3,4] ✉

Direct measurement of neural activity in freely moving animals is essential for understanding how the brain controls and represents behaviors. Genetically encoded calcium indicators report neural activity as changes in fluorescence intensity, but brain motion confounds quantitative measurement of fluorescence. Translation, rotation, and deformation of the brain and the movements of intervening scattering or autofluorescent tissue all alter the amount of fluorescent light captured by a microscope. Compared to single-photon approaches, two-photon microscopy is less sensitive to scattering and off-target fluorescence, but more sensitive to motion, and two photon imaging has always required anchoring the microscope to the brain. We developed a closed-loop resonant axial-scanning high-speed two-photon (CRASH2p) microscope for real-time 3D motion correction in unrestrained animals, without implantation of reference markers. We complemented CRASH2p with a 'Pong' scanning strategy and a multi-stage registration pipeline. We performed volumetric ratiometrically corrected functional imaging in the CNS of freely moving *Drosophila* larvae and discovered previously unknown neural correlates of behavior.

Optical monitoring of neural activity via genetically encoded calcium and voltage indicators has allowed the revolutionary ability to record from populations of identified neurons with cellular resolution[1–5]. Fluorescence microscopy requires comparatively large optical components, of at least mm scale, and is sensitive to changes in the geometry or composition of the optical path between the objective and fluorophore. Even micron-scale motion transverse to the focal plane can cause changes in recovered fluorescence greater than the dynamic range of the indicator. While it can be convenient to make optical measurements in animals that have been immobilized or dissected, many important questions, including in action selection, motor control, and proprioception, can only be addressed by measurement of activity in freely behaving animals[6,7].

Although significant progress has been made by the implantation of miniaturized microscopes in larger vertebrates[2,8,9], optical monitoring of neural activity in freely behaving smaller animals, including insects, has been limited. Single-photon recording of neural activity has been accomplished by a variety of methods in freely moving transparent organisms, including zebrafish[10,11], *C. elegans*[12–20], and hydra[21]. These have produced beautiful results, but the techniques have not transferred to freely behaving insect models, including larval *Drosophila*, an important model organism which has proven difficult to functionally image during free behavior.

In behaving insects, single photon recording has been accomplished in the peripheral nervous system of larval *Drosophila*[17,22,23] and, transiently, of artificially driven activity in regions of the CNS without cellular resolution[24]. In adult *Drosophila*, single-photon recordings of clusters of neurons have been made using a low NA tracking epifluorescence microscope[25,26] without cellular resolution. Aggregating the responses of many neurons in a single measurement can increase

[1]Department of Physics, New York University, New York, NY, USA. [2]Department of Molecular Genetics and Cell Biology, University of Chicago, Chicago, IL, USA. [3]Center for Neural Science, New York University, New York, NY, USA. [4]Neuroscience Institute, New York University, New York, NY, USA. [5]These authors contributed equally: Paul McNulty, Rui Wu. ✉e-mail: mmihovil@syr.edu; marc.gershow@nyu.edu

the available signal and reduce sensitivity to motion, but when single or small sets of neurons control behaviors, sensitivity to the fluorescence of single neurons is needed. When multiple classes of spatially proximate neurons are labeled by a single line, when neurons of the same line communicate different information, or when information is encoded in the spatial distribution of activity, recording with cellular resolution is required.

Two-photon microscopy reduces the effects of scattering, is compatible with optogenetic manipulation, and avoids visible excitation light that would evoke visual responses, but thus far has required a physical connection between the microscope and the imaged organism[8,27–35]. Such a connection (Supplementary Discussion) may perturb the behaviors of larger animals, and entirely prevent the motion of smaller ones, like the *Drosophila* larva. We previously developed a two-photon tracking microscope to record activity from individual neurons in unrestrained and freely crawling *Drosophila* larvae[22,36,37], but two-photon recording from populations of neurons or their processes in unrestrained animals has remained elusive.

Two-photon imaging in unrestrained animals is challenging because two-photon microscopy is sensitive to motion. Motion blur is a challenge in both tethered and free animals; raster scanning samples a volume point by point, and movement during sampling leads to distortion as well as over- and undersampling. For unrestrained animals, the possibility of continuous motion in a single direction poses a further challenge. If an animal moves with continuous velocity in an unknown direction, the volume in which the brain could be located grows like the square of time, for motion confined to a plane, or the cube of time for free motion in general. Meanwhile, the time it takes to sample a volume grows linearly with the volume, so it quickly becomes impossible to "keep up." In restrained mice and zebrafish, two-photon imaging with real-time motion correction has been accomplished by tracking implanted beads with an acousto-optic lens[38], but adapting this technique to record from freely moving and deforming brains without the advantage of implanted beads would not be straightforward (Supplemental discussion).

To overcome these challenges, we developed Closed-loop Resonant Axial Scanning High-speed Two-Photon (CRASH2p) microscopy, capable of ratiometrically corrected functional imaging of extended volumes in freely crawling animals. One scan path of a dual-head microscope operates a tracker to follow a reference neuron, while the other images the surrounding volume, using a scanning pattern designed to combat motion artifacts. Feedback from the tracker stabilizes the imaging path with sub-millisecond latency and sub-micrometer accuracy, while a resonant axial lens allows high-speed volumetric recording. Two-color imaging allows for ratiometric correction of artifacts arising from rotation and deformations of the brain. We used this microscope to record from individual central brain command neurons as well as behaviorally correlated spatially extended activity patterns across the ventral nerve cord (VNC, the spinal cord analog) of larval *Drosophila*, revealing previously unknown neural correlates of behavior.

## Results

We developed CRASH2p microscopy to allow recording from populations of neurons in freely crawling *Drosophila* larvae. Genetically identified sets of neurons were labeled with both a green calcium indicator (e.g., GCaMP6f[39]) and a stable red fluorophore (hexameric mCherry[40]), which was essential to measure and correct artifacts associated with movement. To record from unrestrained animals, we tracked a single neuron as a reference and used feedback from the tracker to stablizie imaging of an extended surrounding volume containing cells and processes. We used CRASH2p microscopy to record spatio-temporal activity patterns from segmentally repeated VNC interneurons and from central brain command neurons.

## CRASH2p microscopy

CRASH2p microscopy (Figs. 1a, 6) used two independent scan heads (referred to as "tracking" and "imaging") to couple closed-loop tracking with volumetric imaging and allow recording from volumes of quickly moving tissue. The tracking head tracked a single neuron as a reference for the separate imaging pathway. The imaging head carried out two-color imaging using a new "Pong" scanning strategy suited to functional imaging in a translating, rotating, and deforming brain. A new data processing pipeline sequentially corrected for rigid motion, non-rigid deformations, and intensity fluctuations.

## 3-axis tracking

Key to real-time motion correction is the ability to track a fixed reference point in a moving brain. CRASH2p uses two independent scan heads (tracking and imaging) for excitation and a shared pathway for detection. The tracking head was dedicated to following a single neuron using its red (mCherry) fluorescence. To avoid cross-talk between tracking and functional imaging, tracking used a 1070 nm pulsed laser that excited mCherry fluorescence only.

The tracking head executed our previously demonstrated 3D tracking technique (refs. 36,37, section "Tracking"). The microscope focal spot was scanned in a cylindrical pattern about the center of a tracked neuron by a combination of in-plane deflectors and a resonant TAG (tunable acoustic gradient) lens[41–43] axial scanner. The resulting temporal pattern of photon emission was used to estimate the displacement of the neuron away from the center of the cylinder; sequential estimates were combined using a Kalman filter to track the reference neuron with sub-micrometer accuracy and sub-millisecond latency. The location of the neuron was then used to stabilize volumetric imaging executed by a second imaging scan head. To maintain both the tracked neuron and the imaged volume within the scanning range of the microscope, the position of a 3-axis stage was continuously adjusted to bring the tracked neuron to the center of the microscope's field of view.

## Two-color volumetric imaging

Given a tracked neuron, we next needed to image a volume surrounding that neuron. The x- and y-galvos of the scan head were driven in a novel "pong" scan pattern designed to enhance later image registration and reduce the effects of temporal correlations in sampling and reconstructing activity patterns (Methods section "Pong Scan"). The position of the tracked neuron was added to the control signal of the galvos to keep the imaged volume centered (in x and y) on the tracked neuron. Axial positioning feedback was provided by adjusting the objective piezo positioner to keep the tracked neuron at the natural focus of the objective. On the tracking head, a TAG lens created a resonant axial scan to enable volumetric imaging.

Unrestrained motion causes rotations and deformations of the brain with spatially varying effects on the fluorescent images that can not be compensated for by measurement at a single reference point. We required simultaneous dual-color volumetric imaging of a green calcium indicator and a red calcium-insensitive fluorophore for accurate measurement of neural activity. To excite green fluorophores, we used a 920 nm laser on the imaging path only, combined with a delayed portion of the 1070 nm laser also used on the tracking head. All green photons collected from the sample were assigned to the imaging head, while red photons were sorted based on a temporal demultiplexing strategy ( Methods section "Temporal Multiplexing").

To demonstrate the advantages of real-time motion correction, the bottom panels of Fig. 1 show the measured red and green fluorescence recorded simultaneously by the imaging head from a crawling *Drosophila* larva expressing mCherry and GCaMP6f in segmentally repeated interneurons throughout the VNC (under *R36G02* control, discussed later). During this 200 ms exposure, the brain translated 100 μm at a top speed of 0.8 mm/s. The left panels (b, c) show projections reconstructed without motion correction, while the right (d, e)

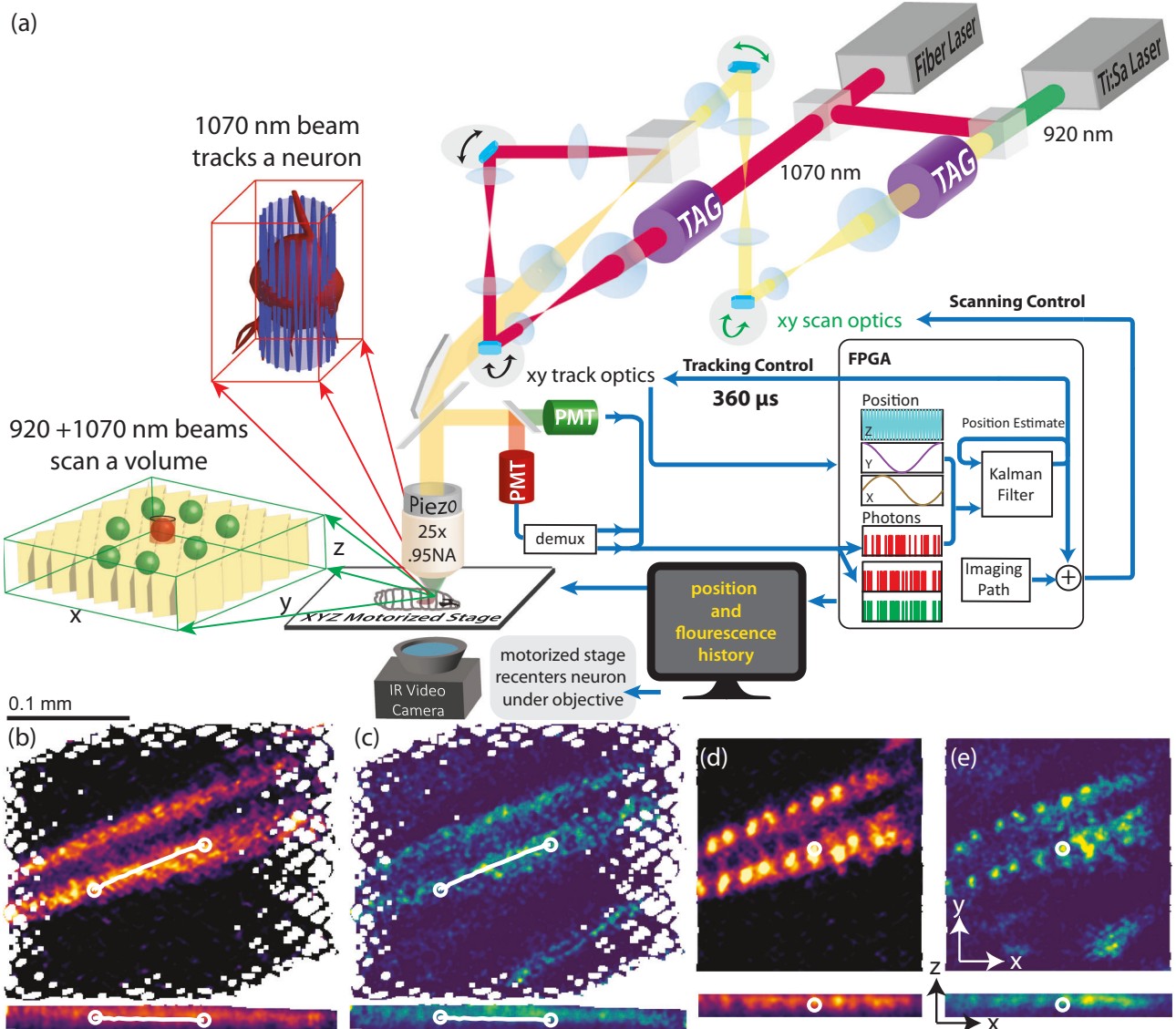

**Fig. 1 | CRASH2p microscopy.** Top (**a**): Principle of operation. Tracking path: a pulsed 1070 nm laser was directed through a resonant TAG lens and relayed on to one scan head's galvanometers. Imaging path: a pulsed 920 nm laser was combined with the 1070 nm laser, directed through a separate tag lens, and relayed on to a second scan head's galvanometers. The 1070 nm pulses were displaced by 1/2 cycle using an adjustable delay line (Fig. 6). The two scan paths were combined and directed through a tube lens onto a 25× objective mounted on a piezo scanner. Fluorescent red and green photons were separately collected by PMTs operated in photon counting mode. The signal from the "red" PMT was temporally demultiplexed; all three (tracking red, imaging red, and imaging green) PMT signals were sent to an FPGA card. Tracking was achieved by rastering the tracking beam in a cylinder about a target neuron; the position of the beam and the record of emitted photons were combined to form an estimate of the neuron's position. Each new

estimate was combined with previous estimates using a Kalman filter and used to direct the next cylindrical scan. The tracked neuron's location was added to the imaging path target location, so the scan was always referenced to the tracked neuron. The piezo positioner provided feedback in z, while a slower motorized stage adjusted all three axes to bring the tracked neuron to the center of the field of view. **Bottom:** Red (**b**, **d**) and green (**c**, **e**) XY and XZ projections from a 200 ms interval without (**b**, **c**) and with (**d**, **e**) tracker correction, during forward crawling. Note that although the tracker output was not used in the assembly of the left images (**b**, **c**), the tracker was still required for their acquisition; otherwise, the labeled cells would have moved outside the imaged volume during scanning. In (**b**, **c**) the path of the tracked neuron is shown with circles indicating the starting (left) and ending (right) positions. In 200 ms, the neuron traveled 97 μm at a top speed of 0.8 mm/s. Genotype: R36G02 > mCherry,GCaMP6f.

panels show the projections with translation corrected by subtracting off the location of the tracked neuron. Although panels b and c were reconstructed as though without the benefit of tracking, in fact, the acquisition of even these blurry images would have been impossible without tracking, as the brain would have exited a 200 × 200 μm planar field of view before the entire image was acquired.

## Correcting for rotations, deformations, and intensity variations

Tracking a reference neuron compensates for translation in real time. However, during free behavior, the brain also rotates about all three

axes, bends, and compresses. While numerous algorithms exist to correct for sample motion, the issues we faced were particularly severe. Unlike for head-fixed or other preparations that constrain the brain to the microscope, the brain could be oriented at any angle in a plane and tilted both side-to-side and front-to-back. Unlike for animals with rigid skulls or for immobilized invertebrates, the larva's brain was deformed during crawling and body bending, with cell bodies displaced many diameters from their unperturbed positions. The efficiency of fluorescence excitation and collection varied with axial position due to imperfections in the microscope, and with posture due

to deformation of labeled cells and changes in intervening scattering tissue, including the cuticle. All of these effects led to spatially inhomogeneous changes in measured fluorescence intensity, even from calcium-insensitive proteins.

To compensate for these effects, we registered the recorded volumetric images to a template using a multi-stage (Methods section "Motion Correction," Fig. S7, Supplementary Movie 12) pipeline. First, we corrected for rigid rotation and residual translation by finding the transform that maximized the correlation between the template and detected photons. Next, we found the best non-rigid registration between the rotated volume and the template using b-spline-based free-form deformation[44]. Finally, we calculated a spatially varying intensity correction using a smooth spline representation to guard against over-fitting. All corrections were calculated using the red (mCherry, non calcium sensitive) channel and then applied equally to both red and green channels.

### In vivo volumetric imaging in moving *Drosophila* larvae

Larval *Drosophila* is an increasingly important model for systems neuroscience[45–48]; its reduced nervous system reflects the architecture of the adult fly's but with fewer neurons and less redundancy; the larva's entire brain has been reconstructed with synaptic resolution using electron microscopy[49–53]; the larva's neurons are genetically identifiable, and a large tool kit exists to label specific neurons, which can be identified in light or electron microscopy[48,54–56]; a comprehensive suite of assays allows quantitative analysis of a range of behaviors[57–63]; the larva is visually insensitive to red light, and optogenetic manipulation of specific neurons in freely moving animals is straightforward[52,64–69] due to the clarity of the larva's cuticle and the specificity of available driver lines. However, despite all these advantages, due to motion and deformation induced by the larva's peristaltic crawling[70], it is exceedingly difficult to measure neural activity in larvae that have not been dissected or fully immobilized. While the *Drosophila* larva is an excellent model for understanding the development and function of motor circuits[46,71], the inability to visualize neural activity in behaving animals hinders progress.

To demonstrate the utility of our microscope, we focused our attention on segmentally repeated interneurons in the *Drosophila* larva VNC, choosing premotor and proprioceptive interneurons. The control of motion and the sensing of posture represent tasks that can only be studied in larvae that are free to move. Understanding these neurons' spatially distributed patterns of activity requires recording at high spatial resolution (for example, if activity in a neuron represents contraction of a corresponding body segment, the sum of all neurons' activities is much less informative than a segment-by-segment breakdown). Finally, the neurons we chose were known or suspected to play a role in forward or backward crawling, a repeated behavior that is easy to characterize and quantify.

For all experiments, we recorded from larvae freely crawling on an agarose gel adhered to a coverslip beneath the microscope objective, i.e., with the ventral surface oriented towards the objective. A larva was placed on the agarose-coated coverslip using a wet paintbrush. The coverslip was then positioned on a holder mounted to the stage underneath the objective, and the larva was immobilized using gentle compression (Methods section "Immobilization stage"). The neurons of interest were located using two-photon microscopy without tracking. A cell body was selected for tracking, the microscope was placed in dual tracking-imaging mode, and the compression was released to allow the larva to crawl freely. While crawling, the larva's behavior and neural activity were simultaneously monitored using IR video recording and CRASH2p microscopy, respectively. This process typically took under 5 min from selection of the larva to recording during free behavior.

### Behaviorally correlated activity waves in segmentally repeated VNC interneurons

We began our investigation with an ideal test case for recording in freely crawling animals: recording the distributed activity of A27h premotor interneurons[46,72,73] during forward and backward crawling. During both fictive[72] and natural[36,37] forward crawling, A27h activity propagates segmentally from posterior to anterior, but A27h neurons are silent during fictive and natural reverse crawling. The expected neural correlates of behavior—forward propagating waves during forward crawling and silence during backward crawling—are unlikely to result from motion artifacts; for most of the peristaltic cycle, the brain is relatively still[36,37,70], and waves of activity that progress during periods of little brain movement are difficult to explain as motion artifacts, nor are motion artifacts likely to be sensitive to whether a larva is crawling forward or backward.

### A27h processes show posterior-anterior activity waves during forward, but not backward, crawling

We labeled A27h neurons with GCaMP6f and hexameric mCherry using the *R36G02-Gal4* promoter line. This line is often termed A27h-Gal4, although other cells are also labeled, including A03g, which is believed to be active in both forward and backward crawling[74]. A03g cell bodies are located in a more ventral plane of the VNC than A27h and can be excluded by restriction of the recorded or analyzed volume to the A27h plane. *R36G02-GAL4* also labels a set of neurons whose cell bodies touch those of A27h, recently termed "M" neurons[75]. The role of M neurons in forward or backward crawling has not been characterized. Of the neurons labeled by R36G02, the processes of A27h can be distinguished by their unique morphology; intrasegmental projections that cross the midline[53,72].

Supplementary Movie 1 shows projections of the registered volumetric images of the VNC of a larva expressing GCaMP6f and mCherry under *R36G02* control, along with infrared images of the behaving larva. The total recording shown spans 253 s, during which the larva crawled a path length of ~1 cm, and exhibited multiple behaviors including forward crawling, backward crawling, and body bending. As expected, when the larva crawled forward, posterior-anterior waves of activity were visible in the green (calcium-sensitive GCaMP6f) but not the red (calcium-insensitive mCherry) channel. Inspection by eye showed that the waves appeared to be synchronized with a contractile wave visible in the infrared image on the body wall and continued even during periods when the brain itself was relatively stationary. These waves were also visible in projections from the unregistered data (Supplementary Movie 2) as acquired by the tracking microscope, which we present to show the difficulties overcome by our registration pipeline.

This is a large dataset, consisting of thousands of volumetric green and red images representing the calcium dynamics of dozens of neurons and their processes during multiple behavioral epochs. To begin our analysis, we started with a strikingly clear feature—behaviorally correlated activity waves in the central (A27h) processes. We identified the central processes on the red template volume used for registration (Fig. 2a). All volumes recorded during this experiment were aligned to this template, so the anatomical markers in the template could be used to identify specific regions of the brain throughout the recording. We manually selected seven volumes of interest (VOI) targeting the A27h processes in seven segments (Fig. 2a), and analyzed the activity in each VOI. The ratiometric activity measure (rate of green fluorescence divided by rate of red fluorescence, "Data analysis") for each VOI over the entire experiment is shown in the stacked plots of Fig. 2b. The behavioral state of the animal is indicated by shaded boxes beneath the traces; at this level of detail, it is already clear that for each VOI, there was more activity measured during forward crawling than during backward crawling, turns or pauses.

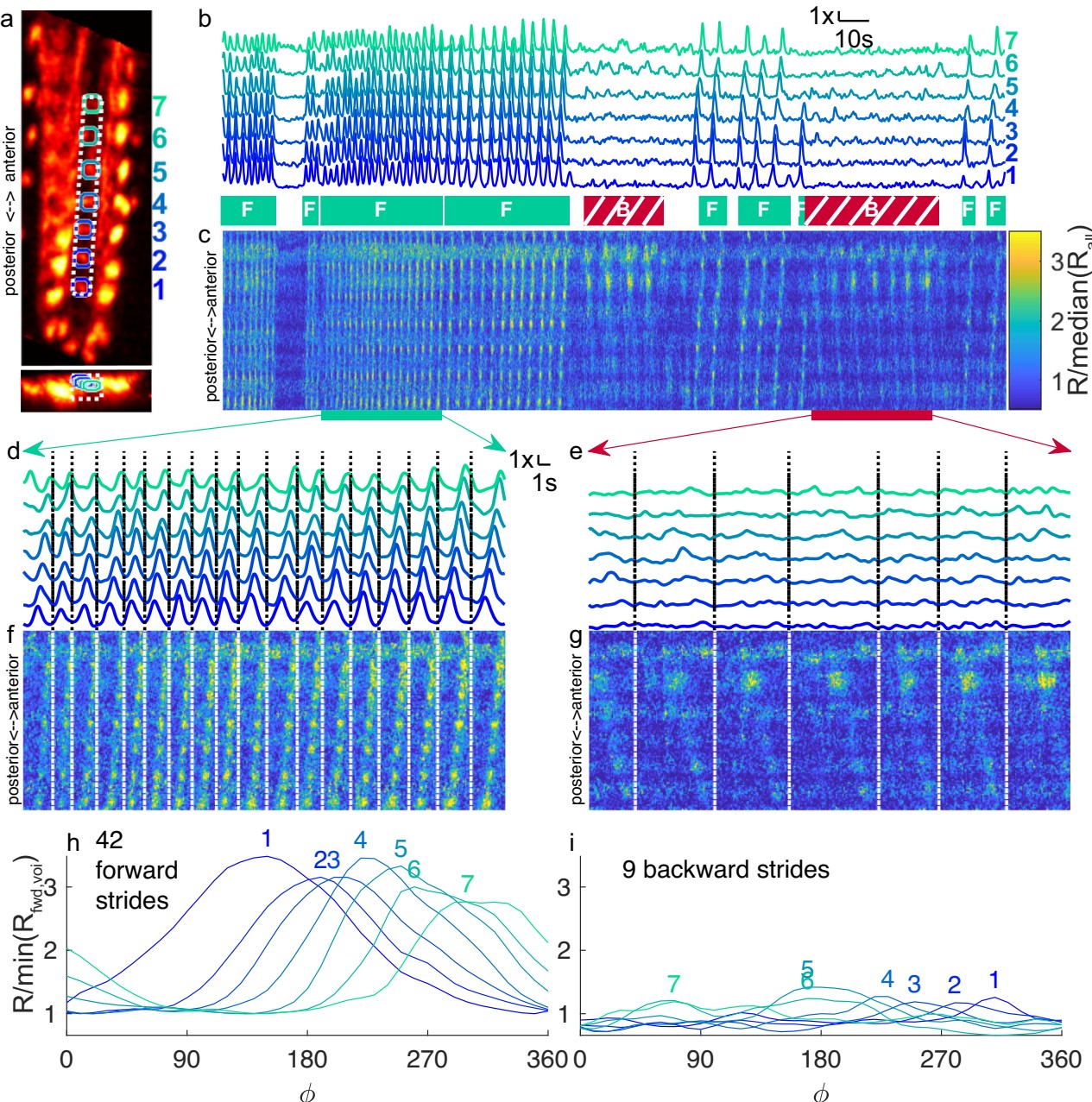

**Fig. 2 | Calcium imaging of A27h central processes.** 2nd instar larva expressed GCaMP6f and hexameric mCherry under *R36G02* control. During the 253 s recording shown, the larva traveled 5.3 mm from its original position along a path length of 1 cm (**a**) X-Y and X-Z projection of template image created from the mCherry channel of labeled VNC. Colored circles indicate VOIs analyzed in (**b**, **d**, **e**, **h**, **i**). VOIs progress numerically from posterior to anterior. The white dashed line indicates the volume projected in (**c**, **f**, **g**). **b** Ratiometric (GCaMP6f/mCherry) measurement of activity in VOIs over the time course of the experiment, normalized separately for each VOI. Periods of forward (teal box labeled with F) and backward (red hatched box labeled with B) crawling are indicated below. Unlabeled

periods represent turns and pauses. **c** Time-space projection of ratiometric activity measure. The vertical (space) axis spans the range from the bottom of the white box to the top. The horizontal (time) axis matches (**b**). The color scale is normalized so that the median of the entire data set shown in the panel is 1. Expanded views of VOI ratio (**d**, **e**) and projected ratios (**f**, **g**) during forward (**d**, **f**) and backward (**e**, **g**) crawling. **c**, **f**, **g** use the same color scale. Cycle-average ratiometric activity measure in each VOI, for forward (**h**) and backward (**i**) crawling vs. cycle phase ($\phi$). Each VOI is normalized separately so that the minimum mean activity during forward crawling is 1.

A potential weakness of the VOI analysis is that hand-selection of volumes might introduce experimenter bias, so we sought a complementary approach to quantifying activity that required less manual intervention. We created a time-space projection, beginning with a box along the centerline of the VNC and summing over the short dimensions of the box, for green and red fluorescence separately. The results were two 2D images in which each pixel represented the integrated green or red fluorescence over a planar slice of the rectangle at a

particular point in time. The ratio of these two images shown in Fig. 2c provides a spatio-temporal map of activity within the center of the VNC. The vertical axis represents space and progresses from posterior at the bottom to anterior at the top. The horizontal axis represents time. In this map, a wave of activity will appear as a tilted bright line. At this level of detail, it is again clear that there is greater activity during forward crawling (positively sloped lines showing posterior-anterior waves) than backward crawling or pauses, but some activity

(negatively sloped lines showing anterior-posterior waves) is also visible during backward crawling.

Crawling consists of a series of repeating strides, and we expect a pattern of A27h activity that repeats with each stride. Figure 2d–g expand panels b, c to allow examination of individual strides for shorter periods of forward (d, f) and backward (e, g) crawling. Dashed vertical lines separate individual strides, determined by analysis of the brain's motion[36,37,70]. During forward crawling, a single posterior-anterior wave can be seen to be associated with each stride, in both the VOI (Fig. 2d) traces and time-space projections (Fig. 2f); during backward crawling, an anterior-posterior wave is visible with each stride in the time-space projections (Fig. 2g) but is not as obvious in the VOI traces.

We asked whether we could determine a characteristic pattern of activity associated with a forward or backward stride. Each stride is of a different duration, and we reasoned that A27h activity is more likely synchronized to the peristaltic cycle than to a fixed temporal clock, so we first defined a new phase "clock" based on the motion of the brain (Methods section "Stride alignment"). A phase of 0° represents the time during a stride at which the brain was moving in the direction of travel the fastest; 360° is the maximum of the next stride. We aligned the ratiometric activity measure of each VOI during each stride to this clock and averaged. Figure 2h shows these average traces for each VOI during forward crawling; each VOI achieved a nearly 3× change in ratio over the course of the cycle and the peak of activity (indicated by the position of the corresponding number) progressed forward in time from posterior VOIs to anterior. During backward crawling (Fig. 2i), the total activity is suppressed (for each VOI, the same normalization is used in panels h and I), but a progression of smaller peaks is visible, this time with more posterior VOIs peaking later in the cycle than more anterior ones.

To show that the ratiometric activity waves are due mainly to changes in green (calcium-sensitive GCaMP6f) and not red (calcium-insensitive mCherry) fluorescence, we repeated the same analysis using the red fluorescence intensity rather than the green/red ratio (Fig. S1). To avoid the possibility of the intensity correction (Methods section "Intensity correction") suppressing apparent waves in the red channel, we conducted the analysis on the red signal without intensity correction. There was minimal variation in the intensity of red fluorescence in each VOI (Fig. S1b, d, e, h, i), while the time-space projections (Fig. S1c, f, g) showed mainly horizontal bands, reflecting the spatial structure of the ladder-like central projections with minimal temporal variation.

## M neurons are active during backward crawling

The complex expression pattern of the *R36G02-GAL4* line allowed us to further demonstrate the utility of CRASH2p microscopy. Even sparse lines may label multiple classes of neurons in close proximity to each other; maintaining high spatial resolution despite rapid brain motion is required to distinguish them from each other. The ability to isolate and interpret spatially segregated signals within the volume is key for the general utility of this scope.

Returning to the entire movie (1), we saw that while the central processes showed the pattern of activity expected of A27h posterior-anterior waves of activity during forward crawling and suppressed activity during backward crawling—the full VNC contained other activity patterns. In the lateral cell bodies, we observed waves propagating from posterior to anterior during forward crawling but also from anterior to posterior during backward crawling. As we did not expect A27h neurons to be active during backwards crawling, we wondered if other labeled cells might be responsible.

We sought a method to map the VNC into regions associated with forward activity waves, backward activity waves, both or neither. If different neurons were active during forward and backward crawling, we would expect this map to show nonoverlapping regions associated

with the different behavioral states. To map out how each spatial point on the VNC contributed to an activity wave during forward and backward crawling, we turned to well-known properties of traveling waves and principal component analysis (PCA).

First, we calculated the stride-aligned axially projected mean green fluorescence intensity for forward and backward crawling separately (Fig. 3a, Supplementary Movie 3). These represent the average spatiotemporal signals associated with forward and backward crawling and show a single posterior-anterior or anterior-posterior wave, respectively. Each spatial point in the dataset is a 24-element temporal signal representing the average green fluorescence intensity at that point vs. phase in the peristaltic cycle. We examined these signals using PCA.

At every point in space, a traveling wave is the weighted sum of two sinusoidal signals 90° out of phase with each other. PCA on a data set containing a dominant traveling wave will reveal these sinusoids as the first two principal components. We found the first two principal components for the forward and backward stride aligned data sets separately; for each set, these components were sinusoidal and 90° out of phase (Fig. 3d), indicating the presence of a traveling wave.

The projection of the signal at each spatial point onto the two components can be expressed as an amplitude and a phase. The phase is assigned to each point independently of the others, but if the activity is mainly due to a traveling wave, then the phase should increase in the direction of wave propagation. Indeed, during forward crawling, the phase increased from posterior to anterior, while during backward crawling, the phase increased from anterior to posterior (Fig. 3c).

The amplitude associated with each point reveals how much specifically wave-like signal is present at that point. Maps of the amplitudes associated with forward and backward activity waves revealed different and largely nonoverlapping spatial patterns of amplitude associated with the two states (Fig. 3b). These spatial patterns were reminiscent of the morphology of the A27h and M neurons[75]; the forward pattern was more medial and contained processes that crossed the midline, resembling A27h. The backward pattern was more lateral and included an apparent ascending process, resembling M. Besides A27h and M, *R36G02-GAL4* labels A03g, whose cell bodies lie in a more ventral plane not imaged here, and which we do not believe to be responsible for the activity measured here.

The M neuron's connectivity has recently been described[76] using the L1 synaptic wiring diagram[53] and confirmed using trans-synaptic labeling. M, identified as A19f, receives input from A18b, which is directly downstream of Mooncrawler (MDN), a backwards command neuron. A18b is already known to be active during backward locomotion[77]. Thus, the two sibling neurons A27h and M[75] appear to have similar roles in opposing behavioral modalities.

M/A19f and A27h are electrically coupled during development[75], but we find them to be oppositely active in the 2nd instar stage. One possible explanation might be simply that the gap junctions present during development are absent in the second instar stage. However, EM reconstruction reveals that in the first instar, A19f directly receives excitatory input from cholinergic A27h[76], so a complete explanation likely requires a fuller examination of the circuit, including local inhibitory interneurons.

## Controls further discount possibility of motion artifacts

We carried out an identical PCA analysis on the red (due to mCherry fluorescence) channel and found no obvious pattern of fluorescence change as a function of peristaltic cycle (Fig. S2a). For neither forward nor backward crawling do the principal components (Fig. S2d) appear to resemble phase-shifted sinusoids, the extracted phases (Fig. S2c) show neither anterior-posterior or posterior-anterior progression, and there is no spatial segregation between the forward and backward crawling amplitude maps.

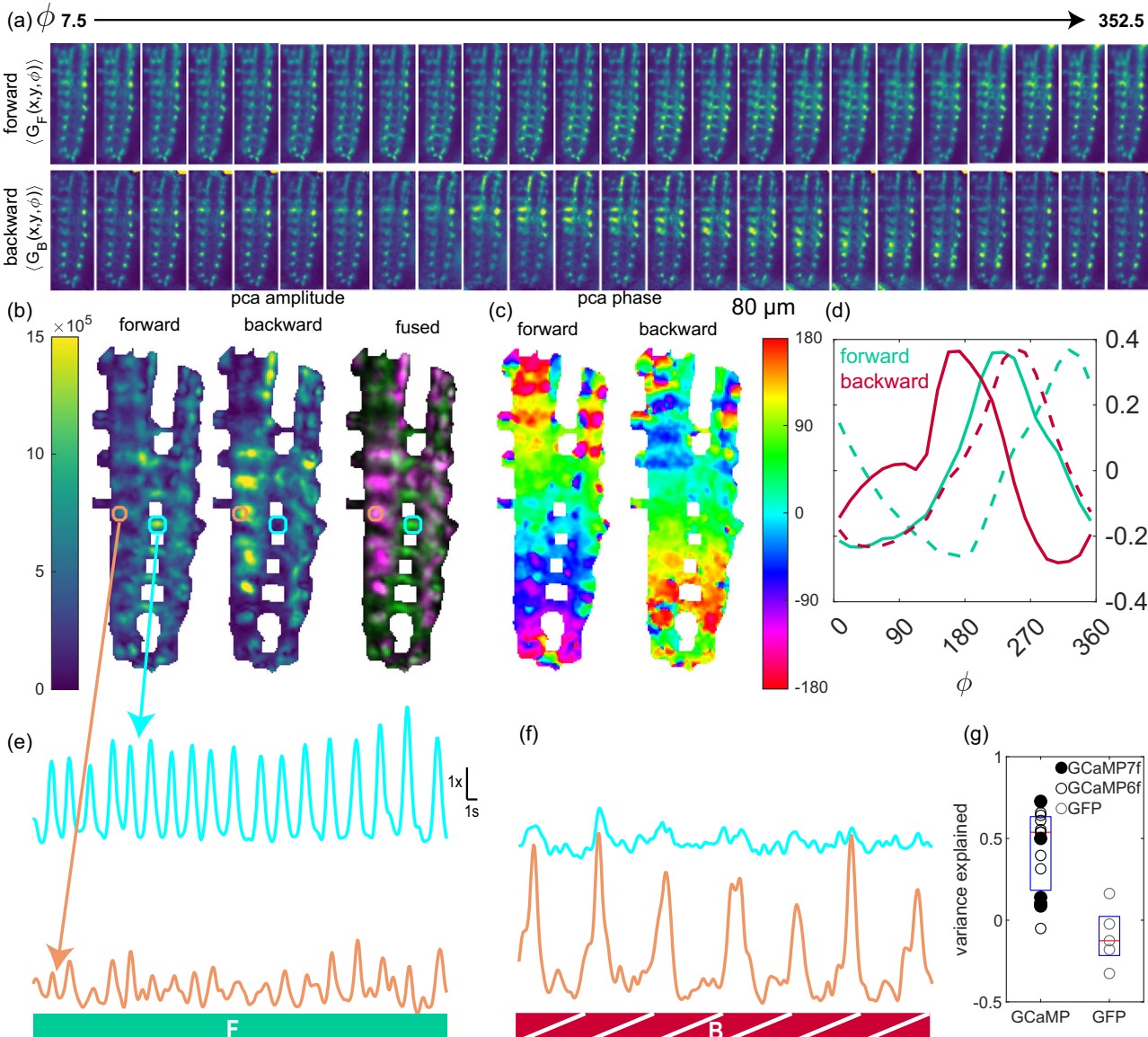

**Fig. 3 | Forward and backward activity waves with distinct anatomical footprints.** Same dataset as Fig. 2. **a** Average green fluorescence vs. position and peristaltic phase ($\phi$) for 42 forward strides (top row) and 9 backward strides (bottom row). Each panel represents 15° of phase or 1/24 cycle. All panels in (**a**–**c**) are 80 μm wide, 222 μm tall, and project over the z- (axial) axis. The color scale is the same in all panels of (**a,b**). **b** Amplitude of projection ($|g(x,y)|$) onto first two principal components during forward and backward crawling. Fused image shows forward projection amplitude in green and backward in purple. **c** Phase of projection onto first two principal components (arg($g(x,y)$)) vs space. The projections were spatially low-passed prior to the calculation of the angle so that regions with low projection amplitude adopt the phase of their neighbors. **d** First two principal components for forward (teal) and backward (red) crawling. Components were ordered so that the first component (solid) leads the second (dashed). Normalized ratio (as in Fig. 2b,d,e) for two VOIs during the forward (**e**) and backward (**f**) crawling epochs highlighted in Fig. 2; the top trace in cyan (same as VOI 4 in Fig. 2) was selected from a region with high PCA amplitude during forward crawling and low during backward crawling; the bottom trace in orange was selected from a region with low PCA amplitude during forward crawling and high PCA amplitude during backward crawling (**g**) Fraction of total variance explained during forward crawling by traveling-wave PCA decomposition, for calcium sensitive and insensitive labeling. The red line is the median, box spans 25th and 75th percentiles. GCaMP6f: 10 animals; jGCaMP7f: 5 animals; GFP 5 animals.

We next sought to understand how these results generalized across animals and indicators. We recorded from additional larvae expressing mCherry along with either GCaMP6f, jGCaMP7f, or GFP under *R36G02-GAL4* control. Unlike for the larva with GCaMP-labeled neurons, in GFP-labeled larvae, we found minimal modulation of the green/red ratio with crawling; neither did we observe a traveling wave-like modulation of intensity associated with forward or backward crawling (Fig. S3, Supplementary Movies 4,5).

To compare across many experiments, we used properties of PCA and waves to create a measure to quantify, without manual intervention, the degree to which each data set contained waves during

crawling. We focused on forward crawling because larvae did not crawl backward in every experiment.

By construction, PCA produces a low-dimensional representation of a data set that best explains the variance of the data set. This representation consists of an amplitude (Fig. 3b) and phase (Fig. 3c) associated at each data point. For a traveling wave, the phase changes linearly along the direction of wave travel, but this is not required by the PCA construction. A wave-like representation with the same amplitude but with the phase fixed to change linearly along a spatial axis will explain variance only if a wave is present in the data set. The larger the fraction of variance explained by the wave-like

representation, the more the activity can be described as resulting from a single traveling wave. In fact, during forward crawling, significantly ($p < 0.01$, Mann-Whitney U-test) more variance of green fluorescence was explained by a traveling wave in GCaMP-labeled animals than in GFP-labeled animals Fig. 3g).

## Spatially distributed neural activity measured with sub-cellular resolution in freely crawling animals

Our *R36G02* measurements showed the following: In larvae labeled with GCaM6f, green fluorescence in the A27h processes progressed in a posterior-anterior wave during forward crawling. Changes in fluorescence were associated with movement of the body even when the brain was still (Supplementary Movie 1). During backward crawling, fluorescence changes in the A27h processes were suppressed. Backward waves were visible in physically distinct (putative M) cell bodies, which were inactive during forward crawling.

None of the above features were observed in the (calcium-insensitive) red channel, nor in the green channel when the same neurons were labeled with (calcium-insensitive) GFP. A reduced-dimensional traveling wave representation explained a substantial fraction of the variance in the green signal, but only when larvae were labeled with calcium indicators. Taken together, these results show that we recorded distributed activity with sub-cellular resolution from the central nervous system of freely crawling larvae.

These results demonstrate the importance of recording from extended volumes with cellular resolution in freely crawling animals. Without the ability to image the entire (or at least a substantial fraction of) the VNC with at least single-segment resolution, we would have been unable to see activity waves. Without cellular resolution, we would have been unable to distinguish between A27h and M neuron activity. And without free behavior, we would have been unable to observe the distinct patterns of activity evoked by forward and backward crawling.

## Even-skipped⁺ lateral (EL) interneurons are active during forward crawling

To further validate the utility of our microscope and to generalize results beyond pre-motor interneurons, we examined activity in Even-Skipped⁺ Lateral (EL) proprioceptive neurons. As for motion coordination, the study of proprioception requires examination of circuit activity in freely behaving animals.

EL neurons are required for bilaterally symmetric motor patterns, and activation of these neurons results in abnormal bending of larvae[24]. While ELs are hypothesized to play a proprioceptive role in the control of forward crawling, recording of EL activity during forward crawling has remained elusive[24]. We recorded from a larva expressing jGCaMP8m[78] and mCherry under control of the EL driver, during an extended period of forward crawling (239 s, 77 strides, 13 mm traveled). We tracked a cluster of EL cell bodies on one side of the VNC but focused our analysis on the central neuropil that consists of processes originating from cell bodies on both sides of the VNC[24]. As for A27h, we observed a posterior-anterior activity wave in the central processes, visible in the registered movie (Supplementary Movie 6), selected vois (Fig. 4b, d), time-space projections (Fig. 4c, e), and stride-aligned average ratios (Fig. 4f), although the magnitude of the changes was less than in A27h, even with the improved 8m indicator. PCA analysis showed posterior-anterior phase progression (Fig. 4g). To confirm these results, we conducted 10 additional experiments, five in larve expressing jGCaMP7f and mCherry under EL control, and five control larvae expressing GFP and mCherry under EL control (Fig. S4). We automatically segmented the VNC based on the rung-like structure (rather than hand-selecting vois) and calculated the stride-aligned average change in green/red ratio. We found that for jGCaMP7f-labeled larvae, each region was active at a peak time that progressed forward in the cycle from posterior to anterior. PCA analysis of the stride-aligned green signal revealed wave-like activity along the ladder rungs, with phase increasing along each segment from posterior to anterior. In contrast, identical analyses of larva labeled with GFP and mCherry showed a smaller modulation of the stride aligned cycle-average ratio and no obvious relation between posterior-anterior position and phase.

We conclude that during forward crawling, at least a subset of the neurons labeled by the EL driver are active in synchrony with the peristaltic cycle, which is consistent with activity in these neurons encoding proprioceptive information useful for the coordination of forward crawling.

## Behaviorally correlated activity of a central brain interneuron

When only one or two neurons are to be interrogated, tracking and recording from individual cell bodies is often sufficient to understand how activity is correlated with behavior. We used our previous microscopes[36,37] to record from central brain interneurons responding to visual stimulus and descending command neurons that evoke backwards crawling. When testing CRASH2p, we chose to focus on measuring spatially distributed activity in both cell bodies and processes— recordings impossible using our previous technology. However, a natural question is whether CRASH2p is also capable of the recordings made previously, of probing the activities of single cells in the central brain.

To test this, we recorded from the Mooncrawler Descending Neuron (MDN), a bilateral pair of command neurons whose optogenetic activation reliably evokes backwards crawling[77]. We previously observed that increased MDN activity predicted backwards crawling[37]; we chose these well-characterized neurons to confirm the ability of CRASH2p to record from central brain cell bodies.

We recorded from eight larvae expressing GCaMP6f and mCherry in MDN neurons and from seven control larvae expressing GFP and mCherry instead. For each larva we tracked the cluster of two contacting cell MDN cell bodies. We recorded from a $150 \times 150 \times 50 \, \mu m$ volume using the tracked cell bodies as a reference. In addition to the tracked cell bodies, the recorded volume included the contralateral cell bodies, mingled processes, and autofluorescent structures, likely including the esophagus, trachea, and cuticle[79,80]. Supplementary Movies 8,9,10,11 show the unregistered volume projections for a larva labeled with GCaMP6f and for a larva labeled with GFP. As the purpose of this investigation was to verify the ability of CRASH2p to record from cell bodies in the CNS, we focused our futher analysis on the tracked cell bodies alone.

To identify the MDN cell bodies, we used a simple off-line tracking algorithm on the red (mCherry) signal only to find the position of the cell body cluster vs. time. We then divided the number of green photons originating from this cluster by the number of red photons to form a ratiometric activity measure. To allow comparison between animals, we normalized each trace to a baseline ratio determined as the average during all periods of forward crawling. Figure 5 (i, ii) shows the time evolution of this measure for eight GCaMP6f-labeled larvae and seven GFP-labeled larvae, shaded by behavioral state. For GCaMP6f larvae only, the green-to-red ratio is elevated during backward crawling (red traces). We further analyzed these traces by considering the average ratiometric activity measure for each "bout" (continuous period) of forward and backward crawling. On an experiment-by-experiment basis, the median of all backward crawling bouts was greater than the median of all forward bouts in seven out of eight animals, with statistical significance of $p < 0.05$ or better in six of these. By contrast, in only 1/7 experiments with GFP-labeled larvae did the median backward bout have a greater green-to-red ratio than the median forward bout (with significance $p < 0.05$). When all experiments of a given genotype were grouped together, the median backward GCaMP6f ratio was significantly ($p = 3 \times 10^{-9}$) greater than the median forward crawling

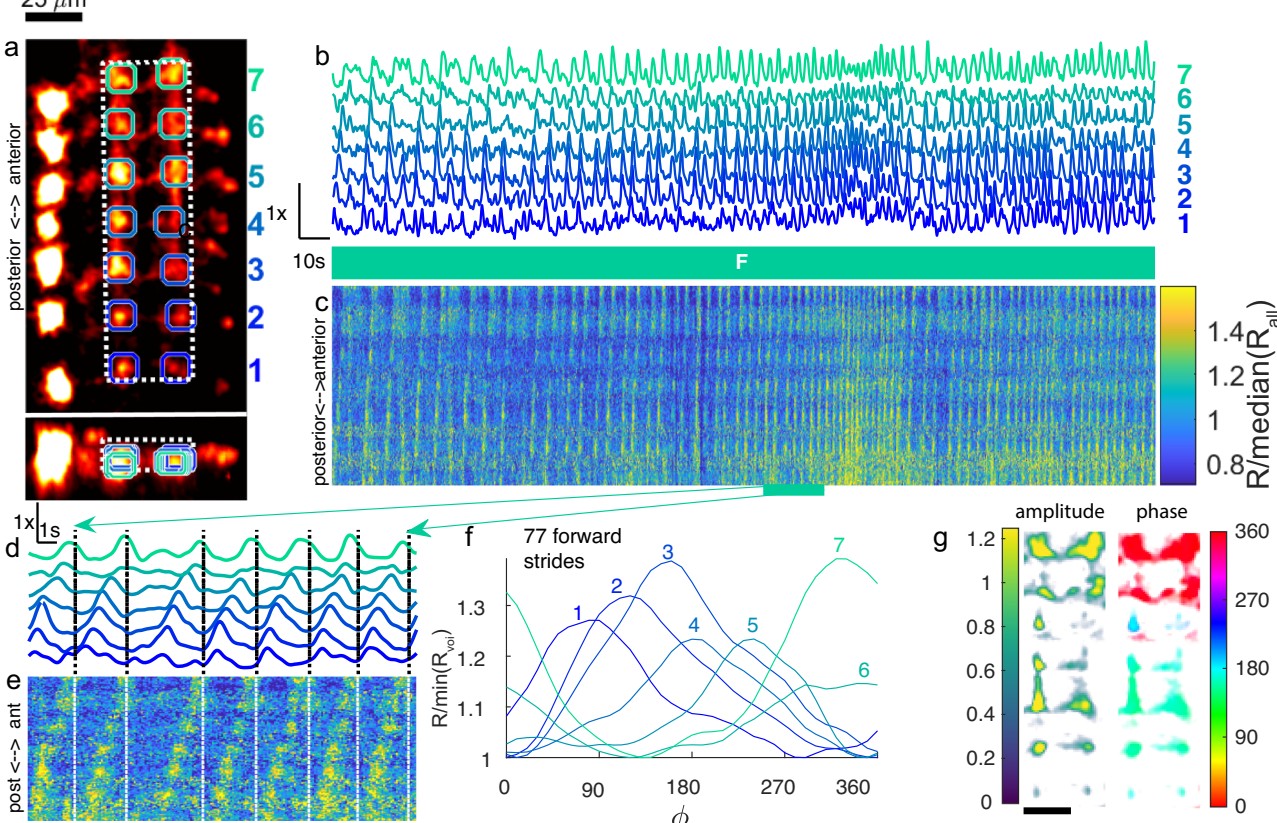

**Fig. 4 | Forward propagating waves in EL central processes.** 2nd instar larva expressed GCaMP8m and hexameric mCherry under EL control. During the 269 s recording shown, the larva traveled 11 mm from its original position along a path length of 13.4 mm (**a**) X-Y and X-Z projection of template image created from mCherry channel of labeled VNC. Colored circles indicate VOIs analyzed in (**b**, **d**, **f**). Each VOI consisted of two spherical regions centered on the nodules on either side of the midline. White dashed line indicates volume projected in (**c**, **e**). **b** Ratiometric (GCaMP8m/mCherry) measurement of activity in VOIs over the time course of the experiment, normalized separately for each VOI. The larva crawled forward for the duration of the experiment, as indicated by the teal box labeled with F. **c** Time-

space projection of ratiometric activity measure. The vertical (space) axis spans the range from the bottom of the white box to the top. The horizontal (time) axis matches (**b**). The color scale is normalized so that the median of the entire data set shown in the panel is 1. (**d**–**e**) Expanded views of voi ratio (**d**) and projected ratios (**e**) during highlighted epoch. **c**, **d** use the same color scale. **f** Cycle-average ratiometric activity measure in each VOI vs. phase ($\phi$) during forward crawling. Each VOI is normalized separately so that the minimum mean activity is 1. **g** Amplitude and phase of posterior-anterior wave in green fluorescence associated with each voxel in the central processes (white boxed region in (**a**)). The amplitude is normalized relative to the median green fluorescence of the region. 25 μm scale bar in (**a**, **g**).

GCaMP6f ratio and significantly different ($p = 8 \times 10^{-7}$) from the GFP backward crawling ratio.

## Discussion

In this work, we demonstrated CRASH2p, a closed-loop volumetric tracking two-photon microscope capable of maintaining micron-scale resolution, even in the face of mm/s scale movements. We used this microscope to record from neurons spanning the entire VNC of a *Drosophila* larva during unrestrained forward and backward crawling. Cellular resolution functional imaging during free behavior allowed us to discover a previously unknown role of M neurons in backward crawling and confirmed a hypothesis that EL neurons should be active during forward crawling.

Larval *Drosophila* is a uniquely powerful insect model for systems neuroscience, with access to the wealth of genetic tools developed for *Drosophila* research, genetically identifiable neurons, a completed EM synaptic wiring diagram, easily interpreted behaviors, and a reduced neuronal count. However, compared to other organisms, progress has been slowed by the difficulty of measuring activity in behaving larvae. Techniques that work well in other models—light sheet, lightfield, and confocal microscopy, or various tethering schemes—have not translated to the larva (Supplementary Discussion). Workarounds, including interpreting motor neuron

activity in dissected or immobilized animals[72,81–83] and photoswitchable activity markers[77,84], have ameliorated but not eliminated this difficulty. CRASH2p allows functional two-photon imaging with ratiometric correction in unrestrained larvae, further advancing the utility of this model organism.

### Imaging in behaving animals

CRASH2p uses our previously described tracking technique[36,37] to provide real-time feedback that stabilizes image formation in behaving animals. In these earlier microscopes, we *recorded* both red and green fluorescence from the tracked cell body, using their ratio as a measure of the Ca++ concentration in that cell body. Using AODs[37], we were able to record from several cell bodies simultaneously. The tracked neurons were continuously sampled at >kHz rates, providing a high-bandwidth and high-fidelity read-out of these neurons' activities. However, these microscopes were unable to form *images* while tracking. CRASH2p was able to form two-color volumetric images extending hundreds of microns, sampled at >10 Hz and with cellular resolution, all while the larva crawled tens of mm at mm/s top speeds. Except for the experiments of Fig. 5, the experiments described in this work, recording simultaneously from dozens of neurons and/or their processes, were beyond the capabilities of those (or any previous) microscopes.

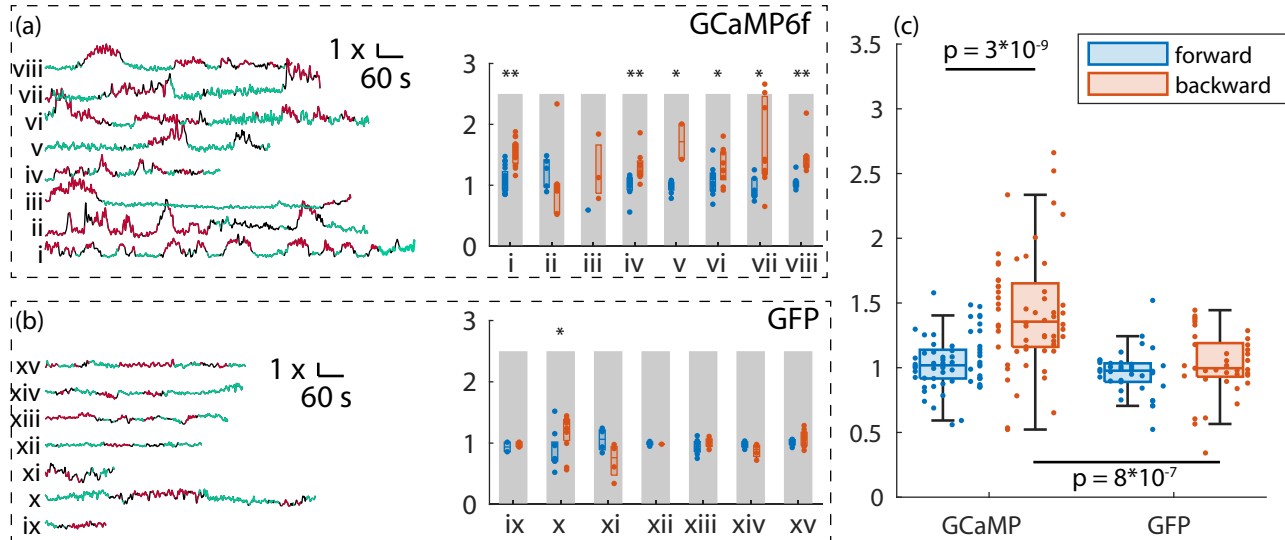

**Fig. 5 | Recording from Mooncrawler Descending Neuron during forward and backward crawling.** One MDN cell body was tracked to stabilize imaging of a larger (150 × 150 × 50 μm) volume. The tracked cell body location was extracted from this larger volume and used to quantify fluorescence ratios. **a** Left: Ratiometric activity measure (green fluorscence/red fluorescence) vs. time for eight different larvae expressing mCherry and GCaMP6f in MDN. Behavioral state (green = forward, red = backward, and black = other) is indicated by color. Each trace is normalized so that the ratiometric measure averaged over all times the larva was crawling forward is 1. Right: Box and whisker plot for each experiment showing the activity measure for each behavioral bout (continuous period of forward or backward crawling) separated by forward and backward crawling. Each data point represents the average ratiometric measure over a single behavioral bout. Central line is the median, box spans 25th–75th percentiles. */** rejects the null-hypothesis that the median of all forward crawling bouts is equal to or greater than the median of all backward crawling bouts (one-sided Wilcoxon rank-sum test) at $p < 0.05/0.01$, respectively ($p$-values: i:7.9e-06, ii:0.84, iii:0.25, iv:0.0047, v:0.022, vi:0.047, vii:0.01, viii:0.0023). **c** Same as **b**, except MDN was labeled with mCherry and Calcium insensitive GFP. ($p$-values ix:0.33, x:0.04, xi:0.94, xii:0.67, xiii:0.18, xiv:0.94, xv:0.47). **c** Box-and-whisker plot of the activity measure of each bout shown in (**a**, **b**) separated by behavioral state and indicator. Each bout is shown as a small dot; bouts from the same experiment are aligned vertically. Central line is the median of all bouts, box spans 25th–75th percentiles, and whiskers span the 5th–95th percentiles. The hypothesis that the GCaMP forward median is greater than or equal to the GCaMP backward median is rejected at $p = 3 \times 10^{-9}$ by the one-sided Wilcoxon rank-sum test. The hypothesis that the GCaMP backward median is the same as the GFP backward median is rejected at $p = 8 \times 10^{-7}$ by the two-sided Wilcoxon rank-sum test.

CRASH2p can be used to record from neurons throughout the larval CNS, including in the brain; our recordings in the VNC most fully demonstrate the advantages of volumetric imaging over tracking alone (Single cell tracking was sufficient to extract the correlation between MDN activity and backwards crawling[37]). Activity in different VNC segments corresponds with different body segments, so a full understanding of how posture is encoded or controlled requires simultaneous recording from many genetically identical neurons across the VNC. In our work with the R36G02 line Figs. 2, 3 recording from processes allowed us to separate the activity of A27h from that of M/A19f, despite their cell bodies touching.

**The challenge of autofluorescence.** One of the challenges in recording from behaving animals is the confounding effect of autofluorescent tissue. If light emerging from autofluorescent structures is interpreted as emerging instead from an imaged cell, then the measured fluorescence of that cell does not accurately report neural activity but will instead reflect a mixture of on- and off-target fluorescence. The off-target element cannot be removed by ratiometric correction.

When recording from stationary tissue, off-target fluorescence due to autofluorescence does not vary in time and the result is an increased level of baseline fluorescence. $F = F_{indicator}(t) + F_{base}$. Given a larger $F_{base}$, $\Delta F/F$ would be smaller in magnitude than without $F_{base}$, but it would still reflect the temporal dynamics of the activity indicator. In behaving animals, however, the amount of autofluorescence mixed into the true activity signal will vary depending on a number of factors that change with time due to tissue movement, including: the amount of excitation light reaching the autofluorescent tissue, compression or deformation of the tissue, the relative positions of the autofluorescent tissue and target cells, and the positions and properties of intervening scatterers. The result is a time-varying $\Delta F$ correlated with behavior that cannot be separated from the neural activity signal.

An advantage of point-scanning two-photon microscopy is a relative immunity to off-target autofluorescence. This is because only one position within the sample is excited at a time and all fluorescent photons detected at that time can be assigned to the particular location of excitation. In moving samples, the signal from autofluorescent tissue can be mixed into the neural activity signal only if motion causes the location of that tissue to be misidentified (i.e., due to imperfections in the image registration process).

The MDN recordings (videos 8–11) show the potential difficulties associated with autofluorescent structures in behaving animals and the power of CRASH2p to overcome these difficulties. The tracked MDN cell body is proximate to a large autofluorescent structure - presumably the esophagus. As the larva crawls, this structure shifts and bends, creating a time- and space-varying fluorescent signal. Despite this, CRASH2p is able to recover the expected behaviorally correlated activity signals from the cell body, taking advantage of point-scanning two-photon microscopy's inherent insensitivity to off-target autofluorescence.

The MDN recordings also show a potential limitation in the use of GFP controls against motion artifacts. The autofluorescent structure is brighter than the GCaMP6f-labeled cell body. While the body is clearly visible in projected views that exclude the autofluorescent structure (Supplementary Movies 9, 8, XY projection), the cell body is completely obscured in projections that also include the autofluorescent structure (Supplementary Movie 8 XZ and ZY projections). In contrast, in the MDN > GFP; mCherry larva, the GFP labeled cell body is much brighter (~10×) than the same structure (Supplementary Movie 10), and

only when the intensity scale is adjusted to a level that over-saturates the GFP labeled cells (Supplementary Movie 11) is the process readily apparent. Assuming the same differences in intensity would prevail in a single-photon recording that is sensitive to scattering, it is entirely plausible that autofluorescence from the esophagus could contribute a confounding signal to a GCaMP recording but barely influence quantification of GFP fluorescence. Thus, while a GFP control can definitely reveal the presence of motion artifacts, it cannot definitively rule out the possibility of motion artifacts due to scattering of background fluorescence.

### Limitations and prospects for improvement

Our microscope accomplishes three functions: tracking a single cell to compensate for translation in real-time, volumetric imaging of a calcium-sensitive indicator for characterization of neural activity, and volumetric imaging of a calcium-insensitive fluorophore for correction of motion-associated changes in fluorescence. The first two functions operate nearly independently of each other and can be optimized separately, but because the same indicator and detector are used for both stable dye imaging and tracking, optimization of one must be sensitive to the needs of the other.

**Closed-loop rotation compensation.** The major disadvantage of our current tracking system is that it compensates only for translation. In addition to requiring more extensive post-processing, untracked rotation of the sample complicates image acquisition. For instance, a particular larval VNC line might best be sampled by a higher aspect ratio rectangular scan pattern (e.g., $200 \times 50\,\mu m$), but because the tracker does not provide real-time feedback on the orientation of the VNC, we must sample a larger square region ($200 \times 200\,\mu m$) to capture all possible orientations. This reduces both the volume rate and the total signal recovered from the neurons of interest. Rotation correction could be accomplished by tracking two neurons using x-y deflecting AODs and a tag lens[37] or with an acousto-optic lens[38,85–88], or perhaps by on-line processing of the reference (mCherry) image.

**Increasing fluorescent signal.** For functional imaging, the operation of the microscope in photon-counting mode limits the total signal available. When tracking single neurons, operation in photon counting mode reduces photo-bleaching and simplifies the FPGA hardware requirements. In the dual-beam microscope, operation in photon-counting mode also simplifies temporal demultiplexing of the red PMT signal. Because our microscope uses an entirely separate excitation laser and PMT for functional imaging than for tracking and ratiometric correction, it would be relatively straightforward to increase the green photon flux available for functional imaging without impacting the tracking functions. Compared to our existing microscope, a wide-bandwidth pre-amplifier and high-speed digitizer would replace the comparator on the output of the green-tuned PMT, and the excitation power delivered by the 920 nm laser would be increased.

Achieving a similar increase in photon flux for imaging the stable indicator would be more complicated because the same excitation laser and PMT are used for both tracking and stable imaging. Unlike for photon counting, time-division demultiplexing of an analog PMT signal requires specialized hardware: high bandwidth amplifiers and fast RF switches or GHz digitizers. Using our current 70 MHz excitation laser, it would be difficult to greatly increase the excitation power on the imaging path without also increasing the power on the tracking path (running the risk of photo-bleaching the tracked neuron), because a certain amount of cross-talk from the two pathways is inevitable. If CRASH2p microscopy were modified to use a lower repetition rate laser (See section "Alternate high speed imaging techniques"), this restriction might be relaxed. In this work, we demonstrated that correction of motion and intensity variation can be achieved in photon-counting mode. Unlike functional imaging, where

increased photon flux would allow higher speed recording from smaller processes, these larger-scale corrections are less likely to benefit from higher photon counts.

**Alternate high-speed imaging techniques.** A variety of recently developed high-speed two-photon techniques[89] allow imaging at frame rates of up to 1 kHz and open up the possibility (not yet realized) of volumetric recording at video-rate. Even at a volume rate of 50 Hz, the motion in this paper would present a significant challenge, with neurons moving up to 10 μm during a single exposure. Depending on the scanning mechanism, movement against the slow axis (Methods section "Pong Scan") could still result in duplicated or missing data. An advantage of our tracking microscope is that the dual-path design means that the tracking mechanism is relatively agnostic to the sampling strategy used on the imaging head. Could we use closed-loop tracking feedback to stabilize other high-speed imaging techniques?

Strategies for faster volumetric imaging generally fall into three categories: faster sampling of single points[41,90,91], computational demultiplexing of multiple points simultaneously sampled by a single detector[92], and camera-based strategies that spatially resolve multiple (quasi-)simultaneously excited points[93,94]. Computational demultiplexing requires a good understanding of the spatial structure of the objects being imaged and thus is unsuitable for moving preparations where this is unknown. It would be complicated to combine tracking with camera-based imaging strategies, due to the different detectors and optical trains required for the two modalities.

Two new fast-scanning strategies use external optical cavities to allow sampling of 30–50 discrete points in a line at a line rate of 1–5 MHz, set by the repetition rate of the excitation laser: FACED[90] samples a line in plane, while light beads microscopy (LBM, ref. 91) samples an axial line. These speeds are comparable to or slightly faster than what can be achieved using a TAG lens (axial line scanning at rates up to 1.1 MHz[41]; in this work we used a 400 kHz line rate both to increase the axial resolution and because the galvo-based pong scanning strategy cannot take advantage of a higher axial line rate). Combining FACED or LBM with closed-loop tracking could have distinct advantages. If only a small axial depth needs to be probed, closed-loop tracking combined with FACED could be used to image a planar region at very high speeds with reduced sensitivity to axial motion. LBM is optimized for sampling discrete planes in thick tissue; replacing the TAG lens on the imaging path with an LBM cavity would allow imaging in thicker samples with closed-loop motion correction.

**Recording from densely labeled tissue.** In this work, we labeled all neurons of interest with both a green calcium indicator and a red stable reference. We tracked a centrally located neuron and imaged the volume around it, The red and green images had identical spatial footprints; the ratio of green to red fluorescence within a volume of interest yielded an activity measure as did the green fluorescence corrected using the red image. This approach requires a spatially sparse labeling scheme—if the labeled cells were to touch without separation, both tracking and image registration would become impossible. To record from densely labeled tissue, we would propose a scheme in which a sparse, distributed set of markers (e.g., neuroblasts, nucleii, glia) is labeled with the stable indicator and the neurons of interest are labeled with a calcium indicator.

We demonstrated two-photon functional recording with cellular resolution in the CNS of a freely behaving, untethered, and unrestrained insect, with ratiometric motion correction. Two-photon recording from any animal not physically anchored to the microscope is challenging, and the larva poses special challenges due to extensive motion-induced translations, rotations, and deformations of the brain. Fast tracking allowed us to compensate for translation, while two-color imaging of a calcium indicator and reference fluorescent protein allowed us to correct for rotation and deformation. The utility

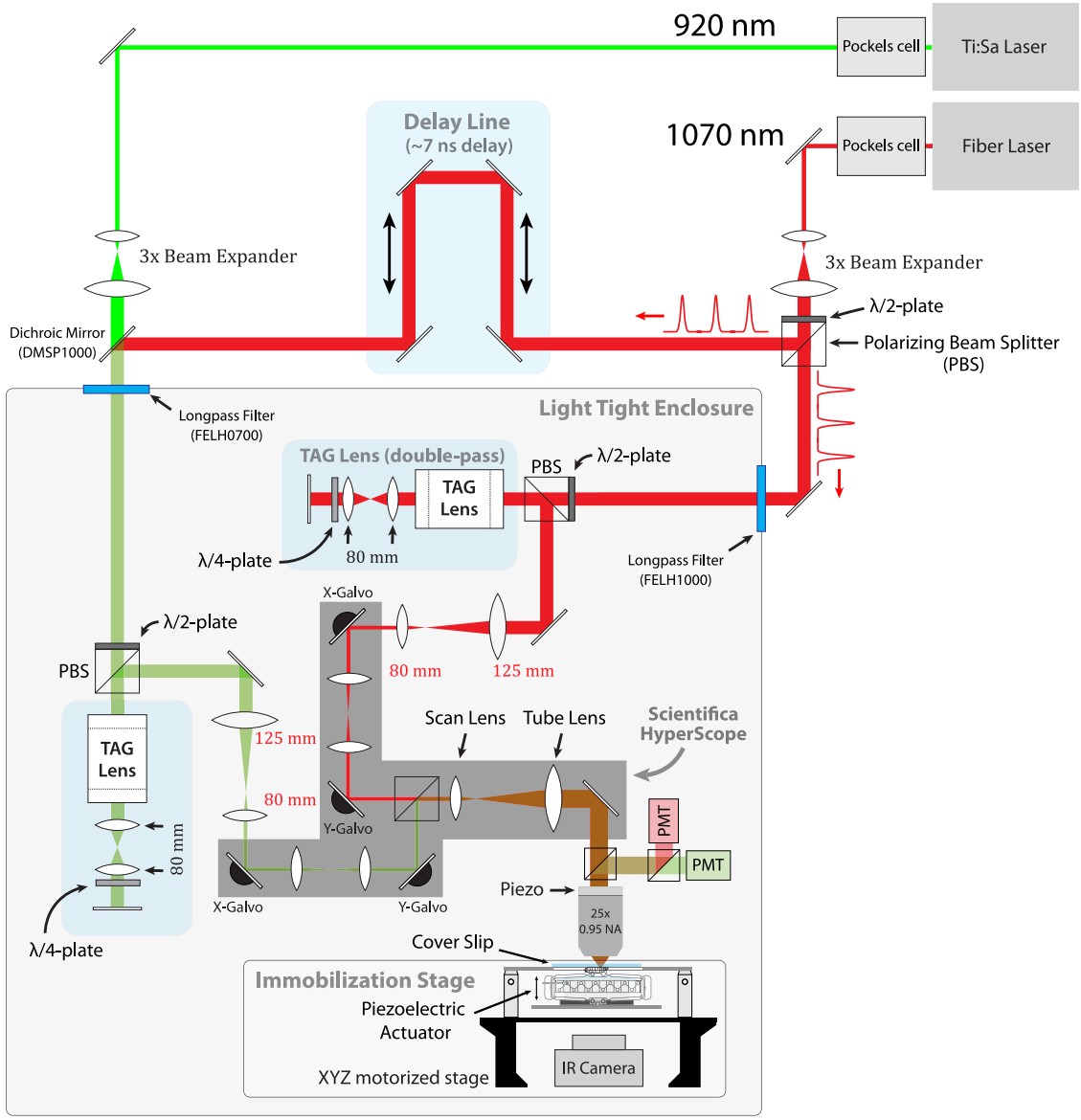

**Fig. 6 | Optical layout of CRASH2p microscope.** The dual-beam hyperscope (shaded gray area in the figure) has two independent scan heads, combined internally with a polarizing beam splitter. One ("tracking") carries a 1070 nm pulsed laser that excites mCherry but not GFP or GCaMP variants; the other ("imaging") carries both 1070 nm and 920 nm pulsed lasers, combined with a dichroic beamsplitter. The 1070 nm laser is split between the two paths, with a delay arm on the tracking path to enable temporal demultiplexing of the photons excited by the independent scan heads. The power of each laser is modulated using a Pockels cell; the ratio of power in the 1070 nm power delivered to the imaging and tracking paths is controlled using a 1/2 wave ($\lambda$/2) plate and polarizing beamsplitter (PBS). Following the pockels cell, the beams are expanded by a factor of 3× to fill the effective aperture of the TAG lenses when operating at 190 kHz. Each path contains a TAG lens in a double-pass configuration. A 1:1.5 demagnifying relay between each TAG lens and the hyperscope conjugates the TAG lens onto an initial blank mirror (place-holder for unused resonant galvo, not shown), the x-galvo, and the y-galvo, which are all placed in the same conjugate plane by means of internal 1:1 relays. The two scan paths are recombined internally by a polarizing beamsplitter and then expanded 6.85× to fill the back aperture of the 8 mm focal length 0.95 NA objective lens. The objective was mounted on a piezo actuator attached to the Hyperscope MDU, which contained filters and optics to separate the IR excitation light from the visible fluorescence emission and direct the latter onto red-tuned and green-tuned PMTs. A 3-axis stage with piezo-electric immobilizer was attached to the table. The larva's behavior was monitored using an infrared (IR) camera mounted below.

of this method was demonstrated by the discovery of a previously unknown role of M neurons in backwards crawling.

## Methods
### Microscope setup
We augmented the dual-scan head Hyperscope (Scientifica, Uckfield, UK) with two ultrasonic acousto-optic lenses (TAG lens, TL25$\beta$.B.NIR, TAG Optics, Princeton, NJ), one on each beam path (Fig. 6). The TAG lenses[41] were used as resonant axial scanners with a resonant frequency of 190 kHz. At this frequency, their effective apertures were 3.4 mm. The

excitation beams were directed through the TAG lenses twice[41] by relaying the principal plane of each lens onto itself by a mirror and a relay composed of two $f = 80$ mm lenses (AC254-080-B-ML).

A 920 nm pulsed excitation laser with $1/e^2$ diameter of 1.2 mm and sech$^2$ pulse width of 140 fs at 80 MHz repetition rate (Chameleon Ultra II, Coherent) excited only the green indicators GFP, GCaMP6f, jGCaMP7f, and jGCaMP8m[40,78,95,96]. A 1070 nm pulsed excitation laser with $1/e^2$ diameter of 1.2 mm and the sech$^2$ pulse width of 55 fs at 70 MHz repetition rate (Fidelity 2, Coherent) excited only the red indicator mCherry[40,95].

The excitation lasers first traveled separately through Pockels cells (1070 through Model 360, 920 through Model 350, Conoptics, Danbury, CT) between crossed polarizers to control beam power. Each laser then passed through a 3× beam expander. The 1070 nm beam was split using a half-wave plate polarizing beam splitter, with roughly half power sent through a 7 ns delay line before being combined with the 920 nm beam path using a dichroic mirror (DMSP1000). Each beam then passed through a longpass filter (FELH0700 for imaging path, FELH1000 for scanning), entering into a custom-built light-tight enclosure that houses the rest of the optical system. The enclosure was designed in AutoCAD 2017 with plugin AutoQouterX® II and the parts were purchased from 80/20® Inc. Inside the box, each beam path had its own TAG lens in a double-pass configuration described above. The beam was then shrunk 1.5× and relayed onto the internal Hyperscope galvanometric scan mirrors. The imaging and scanning paths were then combined and expanded to fill the back aperture of the objective. For all experiments described, a 25×, 0.95 NA water immersion objective was used (HC FLUOTAR 11506375 Leica), mounted on a 100 μm piezo scanner (Nano-F 100s, Mad City Labs, Madison, WI). For the tracking beam, typical power at the back aperture was 30 mW, while for the 1070 nm ("red") imaging beam it was 20 mW and for the 920 nm ("green") imaging beam it was 95 mW.

The objective was mounted on a Scientifica Multiphoton Detection Unit (S-MDU-PMT-45), which contained a dichroic beamsplitter and short pass filter to direct fluoresced photons onto a second dichroic beamsplitter, separating them into red and green channels, each detected by a separate PMT (Scientifica green-shifted S-MDU-PMT-25 and red-shifted S-MDU-PMT-45). The PMT outputs were digitized by Marina Photonics PV16. The PV16 outputs variable pulse lengths. The green photons were regularized to 6.3 ns by a Marina Photonics TTL Oneshot (6.3 ns) while the red pulses were sent to a custom FPGA for de-multiplexing.

The larva crawled on an agarose-coated coverslip held in a custom fixture (See section "Immobilization stage") that was mounted on a 3-axis motor-driven stage (MS-2000 XYZ, Applied Scientific Instrumentation). The XYZ stage received commands from a PID feedback loop running on a Windows PC [64-bit Windows 10 Pro with Intel(R) Core(TM) i7-8700K @ 3.70 GHz and 64 GB memory] to center the neuron to the focus of the objective at 40 Hz (every 25 ms).

Side-looker IR lasers (OPV380, Optek/TT Electronics) mounted on custom PCBs and mounted around three sides of the immobilization stage provided illumination for behavior filming. An IR-sensitive camera (Basler acA640-90um) was set under the stage to record the behavior of the larva through a 1:2.5 demagnifying imaging system composed of $f = 60$ and 150 mm achromats (AC254-060-B-ML and AC254-150-B-ML) and an IR band-pass filter (MV850/40).

The microscope was controlled by custom software written in LabVIEW. Except for the stage, the TAG lens inputs, and the de-multiplexing process, which were addressed by the computer over serial connections, and the immobilization stage which was adjusted manually, microscope hardware (including galvos, objective piezo positioner, green PMT, and TAG lens synchronization signals) was addressed and read out by a multifunction i/o board (NI PXIe-7847) with an integrated FPGA (Xilinx Kintex-7 160T), also programmed with LabVIEW. Real-time functions, including scan generation, tracking, and image assembly, were controlled by the FPGA.

**Agarose coverslips.** An approximately nickel-size droplet of hot 4% agarose (Apex Quick Dissolve LE, Genesee Scientific) gel was dispensed onto a clean 50 × 64 mm No. 1.5 Coverslip (ClariTex). A clean glass slide was placed on top of the droplet and allowed to settle under its own weight for 5 min during which the agarose solidified. The slide was carefully separated from the agarose/coverslip, leaving the agarose attached to the coverslip. Coverslips were adhered to the lid of a petri dish using a droplet of water between the glass side of the slip and the

lid, then stored in the refrigerator inverted (agarose down) over a dish containing DI water to maintain humidity. Before use, the coverslips were placed agarose side up, covered with DI water, and brought to room temperature.

**Immobilization stage.** To smoothly immobilize and release a larva in the beginning of the experiments to select neurons, we designed an immobilization stage (Fig. S8) using two amplified piezo actuators with a travel range of 1150 μm ± 15% (THORLABS APF710 controlled by a K-Cube piezo controller, KPZ101) to control the level of compression. After a larva was placed on an agarose-coated coverslip, the coverslip was placed onto the top of the stage, with the glass surface towards the objective and the larva underneath. Actuating the piezos lifted a piece of clear acrylic upward; to immobilize the larva, the acrylic was lifted until it contacted the larva, as observed by the infrared camera mounted beneath (the camera imaged through the acrylic). The compression was further adjusted during initial imaging to sufficiently immobilize the animal to allow selection of a neuron to be tracked. After the neuron was selected and tracking initiated, the acrylic was lowered well below the point of contact with the larva to allow free crawling.

**Temporal multiplexing.** We labeled cells of interest with two fluorescent proteins: one stable red (hexameric mCherry) and the other green, calcium-sensitive (GCaMP variant) or insensitive control (GFP). To simultaneously image two separate regions with a point scanning multiphoton microscope, the photons from the two regions, which are both directed to the same set of detectors (photomultiplier tubes, PMTs), must be distinguished. We used pulsed lasers of two different central wavelengths: 920 nm, which excited only GFPs or GCaMPs, and 1070 nm, which only excited hexameric mCherry. A dichroic mirror placed after the objective directed red photons to one PMT and green photons to another.

The 920 nm beam was directed only through the imaging path, so green photons could be unambiguously associated with the imaging spot. However, we needed to excite red fluorescence from both scan paths, for tracking the targeted cell and for ratiometric correction of the imaged volume. To distinguish whether photons detected by the red PMT originated from the imaging focal spot or the tracking focal spot, we adopted a temporal multiplexing approach[97–99]. We temporally displaced the tracking and imaging lasers by 1/2 cycle (7.1 ns). The PMT signal was digitized by a fast discriminator and along with the clock signal from the laser, was input to a custom auxiliary FPGA board, which demultiplexed the PMT signal based on the phase of the photon arrival and output the demultiplexed signals to the main FPGA controller. To characterize the cross-talk, we used both scan heads to image the same volume, shuttered the laser on one path at a time, and measured the fraction of photons incorrectly assigned to the shuttered path to be under 10% of the total.

## Pong Scan

To record from a volume, all point scanning microscopes must move the point being scanned in a three-dimensional pattern. A common configuration for two-photon volumetric imaging combines a resonant galvanometer scanner (galvo) for a fast scan in one direction (the *fast axis*) in the objective focal plane and a conventional galvo for a slower scan in the orthogonal planar direction. Together, these allow imaging of a single plane at a rate given by the resonant frequency (e.g., 8 kHz) divided by the number of lines composing the plane (e.g., 256 lines, for a plane rate of 31 Hz). To image a volume, the objective or the sample is translated between each frame, or a remote focusing method (e.g., an electrotunable lens) is used to shift the focal plane of the objective (along the *slow axis*). In this configuration, hundreds of milliseconds can elapse between imaging the top and bottom plane of a sample. Movement perpendicular to the slow axis (in this example, parallel to the focal plane) distorts the image but can be compensated

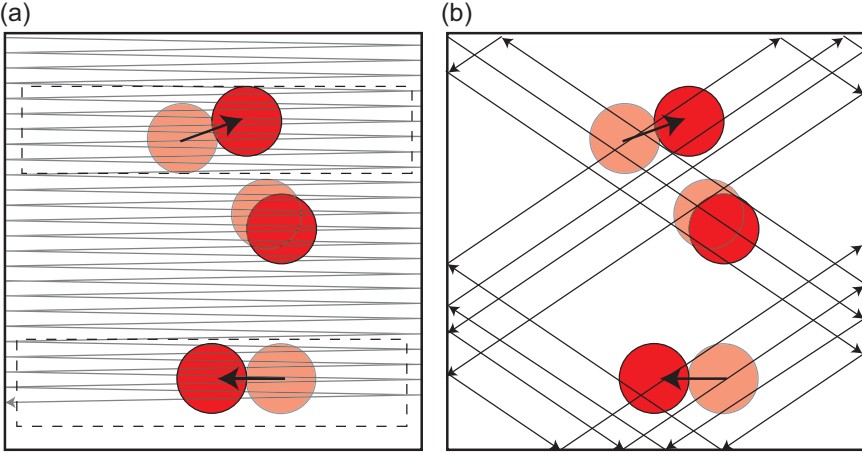

**Fig. 7 | Pong scan vs. raster scan.** Scan patterns overlaying a rotating set of three neurons **a** Conventional raster scan - rotation can appear as lateral movement (dashed boxes show separate times when sample appears to be moving left or right) **b** Pong scan samples all three neurons in rapid succession, making rotation apparent.

in post-processing. Motion parallel to the slow axis can result in over-sampling of some planes and under-sampling of others, which can not be corrected later. Most microscopes image a volume as a series of planes and hence have a slow axis, with light field microscopes and random access microscopes notable exceptions.

Our microscope used a standard galvo-galvo combination for planar (x-y) positioning coupled to a TAG lens for resonant axial (z-) scanning. The z-axis was thus the fast axis, and the resonance frequency (~200 kHz) was fast enough that motion of the sample during a single line scan was negligible (at 1 cm/s, the sample would move only 25 nm in the 2.5 μs required for one direction of the axial scan). At 2.5 μs per axial line, an image consisting of 200 × 200 x-y pixels can be imaged in 100 ms (a volume rate of 10 Hz). In 100 ms, uncorrected motion as slow as 10 μm/s will result in micron scale or larger errors. While the tracker compensated for translation, during body bends, uncompensated rotation resulted in movement on the order of hundreds of μm/s. Given the image of an entire volume, rotations about an axis through that volume are easily identified and corrected. However, if only a partial section of the volume is available, it can be difficult to distinguish rotations from translations and other affine transformations.

To eliminate the slow imaging axis we adopted a strategy related to Lissajous scanning[100–104]. In Lissajous scanning, the x- and y- axes are oscillated sinusoidally at high frequencies chosen to be irrational multiples of each other. The resulting Lissajous pattern fills the imaged plane without repetition. This means that arbitrary resolution in sampling position is achieved given a long enough observation. The trade-off is that for a given desired spatial sampling density, the total imaging time is roughly twice that required for the more efficient raster scan pattern[100,103]. Because the Lissajous pattern repeatedly visits all quadrants of the imaged plane, the data recovered are much better suited to correct for time-varying rotations and other transformations. Like all sinusoidal scans, the Lissajous pattern concentrates sampling time at the border of the image and decreases it in the center. To minimize this effect, we drove the x- and y-axes with *triangular* rather than sinusoidal wave forms, which is possible because we operated the galvos in a non-resonant regime. We named this scan pattern a "Pong Scan" after the pattern of motion of the ball in the 1972 Atari arcade game of the same name.

In addition to motion correction, the Pong scan has advantages for functional imaging. In a conventional raster scan, a cell body is sampled heavily in one time range of each raster and then not at all until the next raster; depending on their positions, different cells are sampled different times. In the Pong scan, each cell is lightly sampled continuously. Because the Pong scan continually fills the plane but samples at increasing spatial resolution with longer times, it is possible

to trade spatial and temporal resolution (e.g., increasing frame rate but decreasing sampling resolution) after the experiment is completed.

Pong scanning is unambiguously superior to Lissajous scanning at the same frequency, because the triangle wave more evenly distributes sampling over time, instead of concentrating the sampling on the periphery of the volume. Lissajous scanning can be used at higher frequencies because resonant scanners, which allow >8 kHz scan frequencies, can only generate sinusoids. Given the need to sample complete axial lines using the resonant tag lens in CRASH2p microscopy, CRASH2p would not benefit from these higher frequencies, and therefore, Pong scanning is superior in this application.

Figure S5 compares a Pong scan to a raster scan for typical parameters used in our experiments: a 200 μm square with a target pixel size of 800 nm (the resonant TAG lens extends this to a 200 × 200 × 50 μm volume). A raster scan strategy completely samples this volume in $\tau_{raster}$ = 189 ms, while a pong scan takes twice as long to completely sample at the same pixel density (Fig. S5a). In 95 ms or $1/2\tau_{raster}$, both the pong scan and raster scan sample roughly half the 800 nm pixels, but the pong scan samples more uniformly across the image (Fig. 2d), and as a result visits every 2 μm pixel in the image at least once during the same time (Fig. S5a, black dashed line). In 0.024 s or $1/8\tau_{raster}$ (Fig. S5c), the raster scan visits only one small band of the image, while the pong scan visits the entire image; this property allows for better correction of rotation (strictly local measurements cannot distinguish translation from rotation—see Fig. 7a). Following a single raster scan, the portion of the volume first sampled by the raster would have grown 'stale' - it has not been sampled in over 100 ms. In contrast, following the same duration pong scan, most 4 μm regions of the plane (columns of the volume) would have been recently sampled (Fig. S5f). This property of repeated sampling, distributed more uniformly over space, provides Pong scanning with an advantage in the reconstruction of time-varying fluorescence traces.

Note that when we discuss sampling a pixel, we mean that we have measured a location within the pixel, but not that we have measured the integrated fluorescence of the entire pixel. Sampling a 2 μm pixel means measuring at least one (PSF-sized) location within that pixel. So it would be possible to miss fine structures within a large pixel in any given sampling pass.

Note also that regardless of scan method, motion associated with behavior makes a priori targeting of sampling locations on micron scales difficult, although the locations can be reconstructed after the fact using our registration pipeline.

To achieve the full benefit of the Pong scan, rather than record images directly from the microscope to disk, we stored the position of

**Table 1 | Imaging parameters**

| Figure | x-freq (Hz) | y-freq (Hz) | x-amplitude (µm) | y-amplitude (µm) | Δx (µm) | $\tau_{raster}$ (s) |
|---|---|---|---|---|---|---|
| Fig. 1b–e, 2,3,S1,S2 | 667 | 412 | 200 | 200 | 0.8 | 0.187 |
| Fig. 4 | 667 | 412 | 200 | 200 | 0.8 | 0.187 |
| Fig. 5 trace o, Supplementary Movie 8 | 889 | 549 | 150 | 150 | 0.8 | 0.106 |
| Fig. 5 trace g, Supplementary Movie 10 | 889 | 549 | 150 | 150 | 0.8 | 0.106 |
| Fig. S4 7f, 32 strides | 1000 | 620 | 120 | 120 | 0.8 | 0.075 |
| Fig. S4 7f, 16 strides | 1000 | 620 | 120 | 120 | 0.8 | 0.075 |
| Fig. S4 7f, 80 strides | 889 | 549 | 150 | 150 | 0.8 | 0.106 |
| Fig. S4 7f, 11 strides | 1000 | 620 | 150 | 150 | 1 | 0.075 |
| Fig. S4 7f, 43 strides | 1000 | 620 | 120 | 120 | 0.8 | 0.075 |
| Fig. S4 GFP, 74 strides | 1000 | 620 | 150 | 150 | 1 | 0.075 |
| Fig. S4 GFP, 28 strides | 667 | 412 | 200 | 200 | 0.8 | 0.187 |
| Fig. S4 GFP, 16 strides | 1000 | 620 | 150 | 150 | 1 | 0.075 |
| Fig. S4 GFP, 98 strides | 667 | 412 | 200 | 200 | 0.8 | 0.187 |
| Fig. S4 GFP, 57 strides | 667 | 412 | 200 | 200 | 0.8 | 0.187 |

Figure: which figure in the text or supplement is referenced—when multiple experiments appear in a figure, these are identified separately by genotype and number of strides; x-freq the frequency of the triangle wave oscillation of the imaging x-galvo (Eq. (1)); y-freq the frequency of the triangle wave oscillation of the imaging y-galvo (Eq. (2)); x-amplitude—the programmed amplitude of the x-galvo triangle wave; y-amplitude the programmed amplitude of the y-galvo triangle wave; Δx: the target pixel size; $\tau_{raster}$, the characteristic time to sample the volume using a bi-directional raster scan.

the beam and the times of photon detection on disk and used these for later image reconstruction[42].

**Imaging settings**

The scan amplitudes and frequencies were determined by the following user inputs: x scan range, $a_x$ (µm), aspect ratio, $r_{xy}$, min dwell time $\tau_{min}$ (µs), target pixel size, $\Delta x$ (µm), and maximum frequency ($f_{max}$). From these, the target x- and y- scan patterns were calculated as follows:

$$x_{freq} = \min\left(\frac{\Delta x}{2a_x\tau_{min}}, f_{max}\right) \quad (1)$$

$$y_{freq} = \frac{2}{1+\sqrt{5}}x_{freq} \quad (2)$$

$$x(t) = a_x \text{tri}(2\pi x_{freq}t) \quad (3)$$

$$y(t) = \frac{a_x}{r_{xy}}\text{tri}(2\pi y_{freq}t) \quad (4)$$

$$\text{tri}(\theta) \equiv \frac{2}{\pi}\arcsin(\sin(\theta)) \quad (5)$$

Table 1 details the specific choices of imaging parameters along with the $\tau_{raster}$, the characteristic raster volume time (Fig. S5) for the corresponding figures.

On the FPGA, the triangle wave was incremented by incrementing or decrementing the target locations by $\Delta x$ at fixed intervals set by an on-board 16 bit counter. The counter values were found by

$$\Delta\tau_x = \frac{\Delta x}{2a_x}\frac{1}{x_{freq}} \quad (6)$$

$$\Delta\tau_y = \Delta\tau_x\frac{1+\sqrt{5}}{2} \quad (7)$$

$$c_x = p_n(f_{fpga}\Delta\tau_x) \quad (8)$$

$$c_y = p_n(f_{fpga}\Delta\tau_y) \quad (9)$$

where $c_x$ and $c_y$ are the counter roll-over values, $f_{fpga} = 80$ MHz is the clock frequency of the loop containing the scan generation code, and $p_n(x)$ denotes the prime number nearest to $x$. Setting the counter values to prime numbers kept the scan pattern from repeating more often than required by the discrete logic of the FPGA.

For registration and display purposes, volumes and their projections were reconstructed using a $1 \times 1 \times 2$ µm x, y, z voxel and discretized into time bins of 1/2 the characteristic raster time (See section "Creating volumetric images"); for the main display figures and movies, this results in a time step of 94 ms.

Temporal signals (e.g. Fig. 2b–e) were lowpassed by convolving with a Gaussian filter of standard deviation of one time step (=1/2 characteristic raster time).

**FPGA code**

We adapted our custom tracking microscope FPGA code[36,37] to this dual-beam layout. The tracking code remained the same; briefly it generated a cylindrical scan of user programmed dimension about the estimated center of the tracked cell; during each approx. 350 µs scan, the originating location of each detected photon was stored, and at the end of the scan, the average location of these photons was, along with previous measurements, used to generate an estimate of the neuron's current location. This location became the center of the next cylindrical scan and was also fed back to the x-, y-, and z-stage and to the objective piezo positioner. This feedback served to keep the tracked cell at the natural focus of the objective lens.

The imaging code, similar to standard 2P imaging software, was adapted from HelioScan[105] with inspiration from PySight[42]; galvos executed a user defined scan path while the time of each photon arrival was recorded by the FPGA and used, along with feedback signals from the galvos and the TAG lens, to assemble real-time image projections; raw data was also streamed to the host computer and recorded on disk for later image reconstruction. During tracking, the FPGA added the estimated location of the tracked cell to the target imaging spot location; thus, the imaged region was always centered on the tracked cell, compensating for translational motion in real time.

**Tracking**

The tracking algorithm described in refs. 36,37 was used with minor modifications. We defined the z-axis of our coordinate system to be parallel to the axis of the objective, meaning the x and y axes were parallel to the focal plane of the objective. To determine each neuron's

location, the galvos directed the focal spot in a circle of radius $R$, typically 3 μm in x and y around the putative center of the neuron, at a frequency of ≤3000 rev/s, the physical limits of the galvos. Coincident with the x-y scan, the TAG lens created a resonant z (axial) oscillation of the focal spot with a tunable peak-peak amplitude of 30 μm and a frequency of 190 kHz.

The arrival time of each photon was recorded during the circle, correlated with the position of the galvos and TAG lens and used to estimate the neuron's location. The TAG scan range extended ±15 μm from the natural focal plane; only photons emitted from within ±$Z$ (typically 5 μm) of the estimated z-location of the neuron were used to estimate the neuron's location. Because we only used the 1070 nm laser for tracking, only red photons were excited on the tracking path and used to calculate the neuron's location. Following each circle, the location of the neuron was updated; full details of the calculation are in ref. 36. Here we reprise the main themes.

With only a single scan, the best estimate of the neuron's location is the center of mass of the emitted photons $\vec{x}_{est} = \frac{1}{N_P}\sum(x_i, y_i, z_i)$, where $N_P$ is the number of detected photons and $(x_i, y_i, z_i)$ is the point of origin of the $i^{th}$ photon. Assuming $R$, $Z$ were chosen appropriately to match the size of the neuron, the error of this estimate, due to shot noise, is approximated by $\sigma_x = \sigma_y = R/\sqrt{N}$ and $\sigma_z = Z/\sqrt{N}$[36].

To combine sequential uncertain measurements, we used a Kalman filter[106–111]. Assume that all errors are Gaussian and following $i$ measurements, the neuron's x-location and uncertainty are given by $x_i \pm \sigma_i$. A time $\Delta t$ later, the neuron's location is measured again. After $\Delta t$ but before applying the measurement, the best estimate of the neurons x-location is unchanged $x_{i+1|i} = x_i$ but the uncertainty has increased $\sigma_{i+1|i}^2 = \sigma_i^2 + D\Delta t$. Note that although $D$ has units of a diffusion constant, because the motion is non-diffusive, $D$ is best understood as a parameter that adjusts the responsiveness of the tracking system. Assume the new measurement places the neuron at $x_m \pm \sigma_m$. We combine the measurement with the previous estimate, weighting them inversely according to their errors. $x_i = (1-K)x_{i+1|i} + Kx_m$, $K = \frac{\sigma_{i+1|i}^2}{\sigma_{i+1|i}^2 + \sigma_m^2}$. The combined error is now $\sigma_i^2 = (1-K)\sigma_{i+1|i}^2 = \frac{\sigma_{i+1|i}^2 \sigma_m^2}{\sigma_{i+1|i}^2 + \sigma_m^2}$. This new estimated location is used as the basis for the next round of measurements.

In this and prior works[36,37], we tracked motion along each axis separately (i.e., the estimate of the y-location did not include any information about the x-location of the neuron or of the emitted photons). While extension to tracking both position and velocity is straightforward[36], in this work we tracked position alone.

**Photobleaching.** In previous work, we observed a decrease in red fluorescence intensity in the tracked neuron due to photobleaching over time; typically, after 15 min of tracking, the red fluorescence rate was reduced to 40% of its initial value[36]. An exponential fit to the red fluorescence rate of the tracked neuron, as measured by the tracker, for the experiment shown in Fig. 2, found that the tracked neuron was reduced to 75% of its initial fluorescence over ~260 s of tracking. For the experiment in Fig. 4, the final neuron's tracked fluorescence was reduced to 80% of the initial. Both of these results are consistent with previously measured rates of bleaching. The MDN experiments (Fig. 5) showed no degradation in red fluorescence over 15 min of tracking, but also recovered lower raw rates of mCherry fluorescence from the tracked neuron. In contrast to our tracking-only microscopes[36,37], where the red fluorescence of the tracked neuron was used for both spatial feedback and ratiometric correction of the green fluorescence signal, in CRASH2p, the tracked neuron's fluorescence is used only for spatial feedback during the actual experiment and not in later analysis.

**Feedback**

Following each measurement, the estimated position of the measured neuron was updated. We used this updated location to calculate the location of the next measurement scan. Other layers of feedback served to keep the tracked neuron centered within the field of view. From fastest to slowest: The x-y galvo deflectors were directed to the estimated location of the tracked neurons (on the imaging path, the estimated location of the tracked neuron was added as an offset to the x- and y- deflection of the scan galvos), the piezo positioner on the objective was set to bring the estimated z-location of the neuron to the natural focus of the objective, and the stage was moved in all three axes to bring the tracked neuron to the natural central position (galvos centered and objective piezo at half range) of the microscope.

Stage position commands were generated by a PID control loop with a 25 ms update rate. Each axis was controlled separately; for all 3 axes the error signal was the displacement of the tracked neuron from the natural objective focus. PID parameters were tuned by adjusting the 100 micron step response of the system to respond quickly without overshooting. Note that the stage controller includes its own PID parameters to translate control signals into motor controls; these were not adjusted.

**Post-experiment image reconstruction and correction**

During experiments, the FPGA assembled XY and XZ projections of the red and green photon counts from the recorded volume in real time; these were displayed on a monitor to provide real-time feedback to the experimenter. The raw data, consisting of the FPGA clock time at which every photon was detected, the FPGA clock time of the synchronizing signals from the two tag lenses, the read-back locations of the galvos and piezo positioner at defined FPGA clock times, and other information used to synchronize the timing of stage readouts and IR ("behavior") camera frames, were streamed to the host computer and written directly to disk[42].

In the initial steps of image assembly, these raw data files were parsed and global corrections applied; the true phases of the tag lenses were determined by matching the distribution of z-locations for photons, and delays in galvo analog feedback were calibrated by maximizing the sharpness of a trial image. Using the data from the imaging scan path, we then began the process of forming images and correcting for motion.

**Axial dwell time correction.** We are generally interested in the *rate* of fluorescence emission, given by the number of photons observed in a voxel or region of interest divided by the time spent probing that voxel/region of interest. If an imaged volume is sampled uniformly, then the raw photon count rate can be taken as proportional to the rate of emission and no further correction is required. Our microscope sampled space unevenly due to the pong scan pattern, the effects of motion, and the resonant axial scan. The first two effects will be discussed later in the methods; correction for the axial resonant scan was as follows:

Due to the resonant axial scan, the time spent within a region $\Delta z$ (the dwell time) varied with phase of the TAG lens ($\phi$). This affects the calculation of the rate vs. axial position.

$$z(\phi) = A\cos(\phi) \tag{10}$$

$$\lambda(z \pm \Delta z/2) = \frac{n(z \pm \Delta z/2)}{\tau(z \pm \Delta z/2)} \tag{11}$$

$$\tau(z \pm \Delta z/2) \approx \frac{\Delta z}{|A\sin(\phi)\omega|} \tag{12}$$

$$\lambda(z) \approx n(z \pm \Delta z/2)\frac{\Delta z}{A\omega}|\sin(\phi)| \tag{13}$$

While it would be possible to apply Eq. (12) directly to calculate the actual dwell time at each voxel, that approach is computationally intensive. So we instead attached the $|sin(\phi)|$ term in Eq. (13) to the *counts* (the numerator of Eq. (11)). We assumed a constant dwell time per voxel equal to the line time ($\frac{1}{2f_{tag}}$) divided by the number of voxels, and applied a correction to the number of observed photons:

$$n_{adj}(z) = n(z)|\sin(\phi)|/0.7846 \qquad (14)$$

where the factor 1/0.7846 was chosen to normalize the total correction to 1 over a typical binning of $\phi$ that excludes the range within $\pm 20°$ of the turnaround points (0° and 180°).

Compared to direct calculation of the dwell time, the adjusted count approach (Eq. (14)) has the additional advantage that the z-line averaged intensity for a scan through a labeled cell is invariant under axial (z-) translation of the cell.

**Determining photon emission locations.** The raw image data consists of a scan trajectory, $\vec{x}(t) = (x(t), y(t), z(t))$, representing the location of the focal spot *relative to the center of the tracked neuron* and the time of photon emissions $t_{photon}$.

$x(t)$ and $y(t)$ were found by subtracting the location of the tracked neuron from the location targeted by the imaging galvos

$$x(t) = x_{galvo}(t) - x_{tracker}(t) \qquad (15)$$

$$y(t) = y_{galvo}(t) - y_{tracker}(t) \qquad (16)$$

$z(t)$ was composed of two parts, the resonant axial scan and an offset reflecting the distance between the tracked neuron and the natural focal plane of the objective

$$z(t) = A\cos(\phi(t)) - z_{off}(t) \qquad (17)$$

Because the tracking microscope continually adjusted the piezo positioner to keep the tracked neuron at the focal plane of the objective, $z_{off}(t)$ was typically less than a micron.

$\vec{x_i}$, the location from which the $i^{th}$ photon was emitted at time $t_i$ was therefore $\vec{x}(t_i)$.

To generate the image projections without tracker correction shown in Fig. 1b, c we used the raw locations without subtracting off the tracker location:

$$x(t) = x_{galvo}(t) \qquad (18)$$

$$y(t) = y_{galvo}(t) \text{ Fig.1b, c only} \qquad (19)$$

$$z(t) = A\cos(\phi(t)) \qquad (20)$$

**Creating volumetric images.** To generate a volume representing a given temporal epoch, we divided a count histogram by a dwell time histogram.

The count histogram was a 3D histogram that binned the locations ($\vec{x_i}$) from which all photons of the desired color (red or green) were emitted during that epoch, following application of the resonant scan adjustment (Eq. (14)).

The linearized dwell time histogram was created by first making a 2D histogram of the (x, y) scan locations at tag phases $\phi = \pm 0°$, i.e., at the center of each axial scan line. This was transformed into a 3D histogram of the same dimensions as the count histogram by assigning the same time per scan line to each voxel, $\tau = 1/(2f_{tag}n_z)$, with $n_z$ the number of z-bins.

Depending on the duration of the time period, the size of the voxels, and deviations of the scanning pattern due to sample motion, some voxels might have been unsampled (zero dwell time). To form template images for registration, we smoothed the raw rate image using L1 spline interpolation[112], which naturally handles missing data.

For movies and other 4D (time + volume) data sets, we constructed 4D histograms of counts and dwell times (following the motion correction steps described below, unless otherwise stated). Each 4D voxel was 1 μm × 1 μm × 2 μm × (1/2$\tau_{raster}$, or 94 ms) in x, y, z, and t, respectively. Unsampled locations had both 0 counts and 0 dwell time. These histograms were convolved with a 4D separable Gaussian filter with a standard deviation of 1 voxel size in each dimension; the rate was found by dividing the convolved count histogram by the convolved dwell time.

**Motion correction**
We used the red (mCherry) data (only) for registration. We first corrected for rigid motion (rotation and translation), then separately for nonrigid deformations (bending, compression, stretching), and finally for variations in fluorescence intensity, due e.g., to motion relative to the natural objective focal plane, compression of the tissue, or occlusion by the cuticle. We then applied these corrections equally to both the green and red channels (Fig. S7, Supplementary Movie 12).

**Rigid registration.** We first created a template using the observed emission rate ($\lambda(\vec{x})$) of photons measured during a 1 s period of minimal motion. We then segmented the observation time into "frames" long enough to oversample the volume by a factor of 3.125 at a x-y pixel density of 1 μm$^{-2}$. For each frame, we found the rigid transformation **R** from the coordinates of the scan path ($\vec{x}(t)$) to the template coordinate system is found that maximized the correlation between detected photons and the template

$$\mathbf{R} = \max_{\mathbf{A}} \sum_{t_{photon}} \lambda(\mathbf{A}\vec{x}(t)) \qquad (21)$$

To assure convergence, we first fit for the overall in-plane rotation angle and a two-dimensional translation vector (three parameters) and then for all six parameters (three rotation angles and a three-dimensional translation vector) of the rigid rotation simultaneously.

During an extended recording, the imaged brain rotates within a fixed rectangular prism imaging volume, and different portions of the brain may be imaged at different times; following initial registration, we combined registered frames from throughout the recording to create a template containing all imaged regions of the brain.

We then carried out a final pass of rigid registration to the combined template at four times higher temporal resolution, with additional continuity terms between the parameters of successive rigid transformations.

**Nonrigid registration.** The rigid registration process corrects for rotation and translation of the brain, but not for plastic deformations, like compression or bending, necessitating non-rigid registration. Beginning with the rigid registration results, we first created a template volume for non-rigid registration by combining several well-registered volumes captured throughout the recording period. We then aligned each rigid-registered frame to the template using the CDMFFD algorithm[44]. Although CDMFFD requires image data as an input, one of the outputs is a vector deformation field $T$ that could be applied to the rigid-registered scan path. Following non-rigid registration of the red image only, we applied the calculated $T$ to the original scan paths of both red and green channels.

**Intensity correction.** In all experiments, we labeled the target neurons with and recorded the fluorescence of a stable red indicator protein

(hexameric mCherry[40]) and a green calcium indicator (GCaMP6f[39], jGCaMP7f[96], or jGCaMP8m[78]) or, for control experiments, a stable green indicator protein (hexameric GFP[40]). Movement of the neuron within the imaged volume, deviation of the neuron from its estimated position, deformation of the brain, and scattering by the cuticle and intervening tissue will all affect the recovered fluorescence. On the imaging path, red and green fluorescence are excited simultaneously at the same focus and recovered through the same objective, and so should be affected equally by these changes.

To correct for motion-induced intensity changes, following non-rigid registration, for each frame $i$, using only the red channel, we found a smooth multiplier, $\alpha_i(\vec{x})$ to the template rate $\lambda(\vec{x})$ that maximized the likelihood of the observed sequence of photon emissions

$$\log(P) = \sum_{t_{photon}} \log(\alpha(\vec{x}_{nr}(t)) - \int \alpha(\vec{x}_{nr}(t))\lambda(\vec{x}_{nr}(t))\mathrm{d}t + const \qquad (22)$$

where $\vec{x}_{nr}(t)$ is the scan path after both rigid and non-rigid registration. To guard against overfitting, $\alpha(\vec{x})$ was formed by cubic b-spline interpolation of control points spaced ~20 μm apart.

Once the correction factor was calculated, it was applied equally to the recovered red and green fluorescence for each frame.

Note that count rates displayed in the figures include this correction and do not directly report the count rate as recorded by the PMTs.

## Data analysis

In this paper, we present three types of images and measures of fluorescence.

1. "Red" represent the fluorescence recovered from the mCherry indicator, after applying the corrections described above.
2. "Green" represent the fluorescence recovered from the GCaMP or GFP indicators, after applying the same corrections derived from the red channel.
3. "Ratiometric" represent the ratio of green to red fluorescence, after applying the corrections to both channels.

**Data representation and manipulation.** For a typical experiment, the registered data set was too large to hold in memory at once. The data was stored on disk in separate files for each small time range, equivalent to the amount of time required to sample the field of view 3.125 times at x-y pixel density of $1\,\mu m^{-2}$ (375 ms for a $200 \times 200\,\mu m$ FOV).

The registered data was aligned to a template whose dimensions were smaller than the entire field of view. Only the portion of the registered data that fell within the volume spanning the template was stored. The total number of red and green photons (counts) and the total sampling time (dwell) were stored as 4D histograms of size NX by NY by NZ by NT, where where NX, NY, NZ were the dimensionless extents of the template image and NT = 4 reflected a discretization of photon arrival and sampling time. These histograms were used as the basis for further computation.

For VOI (e.g., Fig. 2b, d, e, h, i), the rate of (red or green) fluorescence was calculated as the sum of the total adjusted number of (red or green) photons collected from within the VOI divided by the total sampling time. The ratiometric activity measure for a VOI was the green rate divided by the red rate. Because the ratio of green to red fluorescence includes factors (like the overall power of the two excitation lasers and the relative expression levels of the proteins) unrelated to neural activity, we normalized the ratiometric measure through division by a reference ratio, calculated separately for each VOI. In extended traces (e.g., Fig. 2b, d, e), the reference ratio is found by fitting a baseline value by calculating the mean and standard deviation of the data, discarding any values more than 1 standard

deviation above the mean, and then iterating over the remaining data. For stride-aligned data (e.g., Fig. 2h, i), the baseline ratio is the minimum value of the aligned average during forward crawling.

Projections (e.g., Fig. 2c, f, g, Fig. 3a) were calculated as the sum over the projected axis/axes of the adjusted number of photons divided by the sum of the dwell time. The ratiometric activity measure was the green rate divided by the red rate. The baseline ratio was calculated as the mean ratio over the entire data set and used to normalize the entire projected image.

**Posture labeling.** Using the convolutional neural network Social LEAP Estimates Animal Pose (SLEAP)[113], we annotated the infrared videos, labeling the body of the larva using the tip of the head, left and right trachea, a gut feature, and the image artifact caused by the beam scanning spot (Supplementary Movie 14).

**Stride segmentation.** We used the periodic motion of the tracked neuron to establish a clock for stride alignment[36,70]. We began by calculating a smoothed forward speed—the velocity of the tracked neuron in the direction of the head. During forward crawling, the forward speed peaks as the brain moves forward and then remains quiescent. During backward crawling, the brain moves backward and forward in alternation. We first denoised the position estimate by fitting the 2D position of the tracked neuron $x(t), y(t)$ to a smoothing spline using the MATLAB function *csaps* with a smoothing parameter of 0.99. The 2D velocity was calculated by convolving the smoothed neuron track with a derivative-of-gaussian filter ($\sigma = 0.25$s). The smoothed forward velocity, $v_{fwd}$ (Fig. S6a) was the dot product of this 2D velocity with a unit vector parallel to the vector from the SLEAP determined gut location to the SLEAP determined spot location (approximating the location of the brain).

Since $v_{fwd}$ had clear cyclical features representing the crawling strides, we could obtain phase-like information to aid in stride segmentation. For forward crawling, we associated a phase with the velocity using the Hilbert transform H($v(t)$)[114]

$$\hat{v}_{fwd}(t) = H\left(-v_{fwd}(t)\right) \qquad (23)$$

$$v_a(t) = v_{fwd}(t) + i\,\hat{v}_{fwd}(t) \qquad (24)$$

$$= |v_a(t)|e^{i\phi(t)} \qquad (25)$$

Note that we took the Hilbert transform of $-v_{fwd}$ so that $\phi = 0$ corresponded to local maxima of $v_{fwd}$. The $i$th forward stride was defined to span between the $i$th and $(i+1)$th time at which $\phi = 0$. We excluded from the analysis the last stride of each crawling bout.

**Stride alignment.** For each stride, we provisionally used $\phi(t)$ extracted from $v_a(t)$ as the phase within the stride, and defined a quantity $v_i(\phi)$, the instantaneous forward velocity at each stride as a function of phase. We extended the sampling by $\pi/4$ into the previous and next stride to allow for fine alignment of the endpoints; hence, $v_i(\phi)$ was defined for $\phi \in (-\pi/4, 9\pi/4)$ except for the first $\phi \in (0, 9\pi/4)$ and last $\phi \in (-\pi/4, 2\pi)$ stride in a bout.

We then used a dynamic time warping strategy to match each stride to a template. We first defined the template as the average $v(\phi)$ of the strides as determined from the Hilbert transform.

$$v(\bar{\phi}) = 1/N \sum_{i=1}^{N} v_i(\phi); \; \phi \in (0, 2\pi) \qquad (26)$$

We then fit a 4-parameter b-spline phase adjustment $\delta\phi(\phi; \vec{\theta})$ term to each stride's phase to minimize the sum square distance

**Table 2 | Crosses by figure**

| Figure | ♂ | ♀ |
|---|---|---|
| Fig. 3g | R36G02-Gal4 | UAS-GCaMP7f;UAS-6xmCherry |
| Fig. 3g, Fig. S3 | R36G02-Gal4 | UAS-6xGFP;UAS-6xmCherry |
| Fig. 4 | EL-GAL4 | UAS-GCaMP8m;UAS-6xmCherry |
| Fig. 5(i) | MDN-GAL4 | UAS-GCaMP6f;UAS-6xmCherry |
| Fig. 5(ii) | MDN-GAL4 | UAS-6xGFP;UAS-6xmCherry |
| Fig. S4 | EL-GAL4 | UAS-GCaMP7f;UAS-6xmCherry |
| Fig. S4 | EL-GAL4 | UAS-6xGFP;UAS-6xmCherry |

Males of the genotype in the center column (♂) were crossed with virgin females of the genotype in the right column (♀).

between the registered velocity and the mean velocity

$$\theta_{1..4} = \underset{\theta_{1..4}}{\mathrm{argmin}} \int_0^{2\pi} \left( v_i(\phi + \delta\phi(\phi; \overrightarrow{\theta})) - v(\bar{\phi}) \right)^2 d\phi \qquad (27)$$

In addition to enforcing smoothness of $\delta\phi$ through spline representation, we also restricted the maximum shift: $|\delta\phi| < \pi/4$.

Finally, we mapped the adjusted phase $(\phi + \delta\phi)$ of each stride back to the original experimental clock and smoothed out discontinuities at the stride boundaries. This formed the $\phi(t)$ for the forward crawling portion of the data set that was used to align neural recordings to phase (as in Fig. 2h, i, Fig. 3a).

An identical procedure was used to calculate the phase during backward crawling, except we took the Hilbert transform of $+v_{fwd}$ so that $\phi = 0$ corresponded to points of maximum backward velocity.

Figure S6a shows $v_{fwd}$ as a function of time for the experiment shown in Fig. 2. Note that $v_{fwd}$ is calculated from the smoothed and low-passed position—the actual peak velocities of the tracked neuron were much higher than the peak values of $v_{fwd}$. Cyan and red dots indicate the initiation ($\phi = 0$) of a stride. Figure S6b, c shows $v_{fwd}$ for each stride aligned to the phase clock determined following time warping, and the average of all strides, for forward (b) and backward (c) crawling. This same phase clock was used to align the voi signals in Fig. 2h, i and in the PCA analysis of Fig. 3.

**PCA analysis of traveling waves.** We began with the registered and intensity corrected green fluorescent signal $G(x, y, z, t)$ recorded during forward crawling. We summed over the z-axis, aligned each stride to the clock determined by the larva's motion, and calculated the average over all strides to create a 3D (2 spatial + 1 temporal dimension represented as a phase $\phi$ between 0° and 360°) data cube, $\langle G_F(x, y, \phi) \rangle$ (Fig. 3a, Fig. 3). We similarly calculated the phase-aligned green fluorescence during backward crawling ($\langle G_B(x, y, \phi) \rangle$, Fig. 3a, Supplementary Movie 3) and the corresponding values for red fluorescence. ($\langle R_F(x, y, \phi) \rangle$, $\langle R_B(x, y, \phi) \rangle$, Fig. S2a, Supplementary Movie 3).

We then carried out Principal Component Analysis (PCA) on the mean-subtracted temporal signals: $\bar{G}(x, y, \phi) = \langle G(x, y, \phi) \rangle - \bar{G}(x, y)$. If a traveling wave is the dominant feature in a data set, then the first two principal components will be sinusoids 90° out of phase; we ordered the components $c_1(\phi)$ and $c_2(\phi)$ (Fig. 3d), so that $c_1$ led $c_2$, and represented them as a single complex temporal signal $C(\phi) = c_1(\phi) + ic_2(\phi)$. A dimensionally reduced representation of $\bar{G}(x, y, \phi)$ is formed by

$$\bar{G}(x, y, \phi) \approx \Re \left( g^*(x, y) * C(\phi) \right) \qquad (28)$$

At each point $|g(x, y)|$ (Fig. 3b) represents the magnitude of the signal contained in the first two components (i.e., potentially associated with a traveling wave), while $\arg(g(x, y))$ (Fig. 3c) represents the phase up to an additive constant.

To quantify the presence of an activity wave associated with forward crawling, we calculated the fraction of variance explained by a traveling-wave projection onto the first two principal components. As before (Eq. 28), the projection onto the first two components was calculated as a complex function of space, $g(x, y)$. Then, while maintaining the magnitude $|g(x, y)|$, we restricted the phase to that of a traveling wave:

$$g(x, y) = |g(x, y)|e^{i(\overrightarrow{k} \cdot (x, y) + \phi_0)} \qquad (29)$$

$k$ and $\phi_0$ were fit parameters chosen to minimize the remaining variance after subtracting off the projected signal. Note that while by construction, unrestricted projection onto the first two principal components will explain a large fraction of the variance, subtracting off the wave-restricted projection may actually increase the residual variance, in which case the variance explained would be negative. The larger the fraction of variance explained by the wave-like projection, the more the activity can be described as resulting from a single traveling wave.

### Fly husbandry
The following strains were used: R36G02-GAL4(Bloomington 49,939[72,77]), EL-GAL4[24], UAS-6xmCherry[40] (Bloomington 52268), UAS-6xGFP (Bloomington 52261,[40]), UAS-jGCaMP7f (Bloomington 80906,[96]), UAS-GCaMP6f (Bloomington 42747,[39]), UAS-jGCaMP8m (Bloomington 92590,[78]), SS01613-GAL4 (MDN,[77]).

Males of the driver line were crossed against virgin females containing both fluorescent reporters (UAS-6xmCherry on 3, UAS-indicated green indicator on 2, Table 2) and allowed to lay eggs on 60 mm egg collection plates. Larvae were used as second instars (verified by size and spiracle morphology) 48–72 h AEL, without regard to sex, due to the difficulty of determining the sex of second instar larvae from visual examination.

### Reporting summary
Further information on research design is available in the Nature Portfolio Reporting Summary linked to this article.

## Data availability
The data generated in this study have been deposited in the Harvard Dataverse database under accession code https://doi.org/10.7910/DVN/ZNJ8U9. Software to read and analyze these datasets is available on Github at GershowLab/CRASH2p Source data are provided with this paper.

## Code availability
MATLAB software used in this publication is available on Github at GershowLab/CRASH2p.

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

## Acknowledgments

This work was supported by NSF Award 1455015 (M.G.), NIH Award 1DP2EB022359 (M.G.), NIH Award R01-NS105748 (E.S.), and a Sloan Research Fellowship (M.G.).

## Author contributions

Design, construction, and programming of microscope: P.M., A.Y., M.M.S., M.G. Carried out experiments: P.M. Wrote Software: R.W., M.G. Integrated SLEAP Labeling: A.N., R.W., P.M. Analyzed data: R.W., P.M., M.G. Provided Genetic Reagents: E.H. Temporal Demultiplexer: A.H., M.G., A.Y., P.M. Drafted manuscript: P.M., R.W., A.Y., E.H., M.G. Supervised project: M.G.

## Competing interests

This study does not involve experiments involving vertebrate animals, human participants, or clinical samples. The authors declare no competing interests.
