## [Transparent Peer Review file · Nature Communications]

Closed-loop Two Photon Functional Imaging in a Freely Moving Animal

Corresponding Author: Dr Marc Gershow

Version 0:

Reviewer comments:

Reviewer #1

(Remarks to the Author)

Reviewer #2

(Remarks to the Author)

In this manuscript the Authors present the development of a closed-loop resonant axial-scanning high-speed two photon microscope able to acquire nervous activity in real-time and in 3D in unrestrained *Drosophila* larvae. The Authors used this instrument to study in great detail the nervous activity in the ventral nerve cord during strides. The Authors convincingly illustrated the advantages brought by their new technique in the feasibility of this kind of measure. The scientific content is valid and original and it is of interest by the biological and the imaging community. The technique developed is ingenious and demonstrates high technical capabilities. Therefore, I suggest the Editor to consider this manuscript for publication.

There are however few issues that need improvement.

1. It is cumbersome to find in the text information about the field-of-view dimensions (specified in the section 8.8.1) and volumetric imaging frequency. About the last point, the Authors state that pong scan allows to select the tradeoff between spatial resolution and temporal resolution offline, after the experiment. This is an interesting point. Nevertheless, it is cumbersome to find in the text the actual (volumetric) temporal resolution which was used in the final analysis. My advice is to put these information more in evidence in the text, making them more accessible to the reader.
2. The manuscript makes a convincing argument for the validity of the presented method for the imaging of *Drosophila* larva. However, no other animal was tested. Therefore, the reader can't assess the generalizability of this method to image the nervous activity for different animal models. The Authors do not discuss this point adequately. This issue is exacerbated by the manuscript title, which states: "imaging in freely moving animals", suggesting a generalization which is not really present in the text.
3. Time-space projection of ratiometric activity measures are present in several figures of the manuscript, e.g. Fig 2c. These images represent a "carpet" generated by mapping an 1-D average projection of space on the vertical axis and the time on the horizontal axis. This approach was used by the Authors to avoid potential experimenter bias introduced by ROIs drawing. This is actually a good idea. However, these images were not analyzed in a quantitative manner, instead they are only discussed in a qualitative way. This frustrates the advantage of a bias-resistant approach, therefore the Authors should analyze them in a quantitative way. Several different approaches are possible. My suggestion is to perform a 2D power spectrum (i.e. the modulus of the 2D Fourier transform squared) of the "carpet". From the 2D power spectrum it is then possible to retrieve the average inclination and intensity of the "tilted bright lines" for the different conditions. Since each stride is of a different duration, this approach would probably work better if the time on the horizontal axis would be replaced by the phase "clock".
4. The usual DF/F0 transform for the fluorescence traces does not seem to be used in the manuscript. The Author should comment about this choice.
5. Regarding section 8.8.5 "PCA analysis of travelling waves", I identify several issues:
 - a. "If a traveling wave is the dominant feature in a data set, then the first two principal components will be sinusoids 90

degrees out of phase;”

Can the Authors give a justification for this statement? A mathematical tractation or, at least, a reference to another article explaining this relation.

b. Regarding the right side of equation 16: does the “R” symbol means the “real part”? Shouldn't it be instead the modulus of the complex number? Why is the point value z present as a complex conjugate? Isn't it a real number?

c. The last paragraph of this section starts at line 1513 (“To quantify the presence of an activity wave...”). It is not clear to me which part of the “Result” section is referred to by this part of the “Methods” section, maybe the last part of section 2.2.4 (from line 566)? If that is so, then I am not convinced by this method. The fraction of explained variance in the two conditions is compared starting from two models obtained from the independent best fit of two variables. Therefore, the fraction of explained variance now depends both from the starting condition of the two different conditions and from the results of the fitting procedure. Was the second factor adequately controlled by the Authors? It could introduce additional variability in an analysis that is presented as quantitative.

Moreover, this procedure generates mathematically-strange results such as a negative variance: a quantity that is defined as not-negative by definition. Therefore, it is difficult for me to trust such an analysis method without a proper and extensive validation. Nevertheless, is this comparison really necessary? It seems to me an overcomplicated approach to try to quantify a phenomenon that is already qualitatively evident from the generated images (GCaMP vs GFP).

6. Movies M4 and M5 are not explicitly referenced in the main text.

Reviewer #3

(Remarks to the Author)

In this paper, McNulty, et al. present a new microscopy technique, CRASH2p, which enables two photon imaging in moving animals. They demonstrate that their approach allows for the recording of population activity in the ventral nerve cord of *Drosophila* larvae with single neuron resolution. This is an important technique for systems neuroscience because *Drosophila* lead the field in many tools critical for dissecting neural circuits (e.g. single-cell targeting genetic tools, a whole-brain connectome, etc.) yet one important tool lacking in this animal is the ability for researchers to monitor cell-resolved neural activity in unrestrained behaving animals, as is possible in larger animals, for example, the mouse. This paper makes a significant advance towards addressing this significant need. The paper is well written, and their approach has the potential to open new avenues of investigation for matching behavior with its underlying neural origins using larval *Drosophila*. As such, this paper should be of significant interest to the general readership of *Nature Communications*.

I do have several concerns that should be addressed before publication:

1. While impressive, CRASH2p does have limitations. The authors address some of the technical limitations in the discussion, but it would be helpful if they could discuss limitations on potential applications as well. One limitation is that the use cases of CRASH2p appear narrow, as imaging neural activity in small, freely moving, transparent organisms (eg. *C. Elegans*, zebrafish) does not require two photon microscopy, and larger model organisms (eg. mice) can support the weight of miniaturized microscopes. Secondly, CRASH2p does not allow for the imaging of larger or faster insects (for example, adult *Drosophila*), meaning that the use of this technology is limited to *Drosophila* larvae and other small, slow-moving organisms for which one-photon access to the entire nervous system is not possible.
2. CRASH2p is introduced as a technology that allows for functional imaging in the larval CNS, but the proofs of concept provided by the authors are restricted to imaging in the VNC. Is imaging in the brain feasible with CRASH2p? Would it be possible for the authors to provide evidence of successful brain imaging with single neuron resolution?
3. All the examples provided by the authors appear to be imaging from neuronal processes, and not cell bodies. Does CRASH2p have sufficient spatial resolution to identify and image activity in cell bodies? Can cell bodies be spatially resolved from one another?
4. The single tracked neuron appears to be scanned continuously within a small ROI, raising concerns that photobleaching may degrade tracking quality over the course of an experiment. Could the authors provide data that quantifies the photobleaching of the tracked neuron over time? Further, could they quantify the relationship between bleaching and tracking quality to demonstrate that their tracking approach remains reliable? It would also be useful for readers to know at what point, if any, this becomes unreliable.
5. The authors initially expect, given previous work, that A27h pre-motor interneurons will be silent during backward crawling (lines 309-311), but observe a front-to-back wave during backward crawling (Figure 2G, lines 412-414). This effect is much weaker than the back-to-front waves present during forward walking but is still present. They explain this by demonstrating different patterns of spatial activation during forward vs. backwards walking and suggest that the ROIs active during backwards walking are M neurons also labeled by their Gal4 line. While I find their PC analysis (Figure 3) compelling, their findings leave me puzzled as to how the broader circuit may be operating. Zeng et al 2021 (the authors' citation #65) found that A27h and M neurons are electrically coupled via gap junctions and show synchronous activity patterns. This seems incompatible with the authors' finding that A27h and M neurons are differentially recruited by opposing locomotor programs. Can the authors address this discrepancy?
6. The authors' identification of distinct cell types labelled by the R326G02-Gal4 line is based on the relative position of ROIs, which is not the most reliable method for cell type identification (especially when cell bodies and processes are highly overlapping, as appears to be the case for A27h and M neurons in citation #65). Can the authors unambiguously confirm the morphology or identity of recorded neurons (e.g. by comparing the targeted cell morphology to the connectome or other method)? From the connectome, are there any neurons with similar morphologies that the authors may be mistaking as A27h or M neurons? Because of the unexpected result for M neuron activity (see point #2 above), it is particularly important that the identification of these neurons is robust if the authors wish to draw any conclusion about M cell function during crawling.

7. Could the authors add discussion of how their findings relate to the function of the broader circuit during larval locomotion? Their results are interesting but likely difficult for the reader to contextualize with the provided information. Even with the main focus of the paper being the microscopy technology, a broader locomotion circuit discourse does seem lacking given the emphasis on this circuit and the asserted novelty of the findings.

8. The authors use a “pong” scan pattern which they claim improves image registration and reconstruction compared to a raster scan pattern or a Lissajous scan pattern. Given that this scan pattern appears to be a novel approach, can the authors provide quantification of these improvements?

Reviewer #4

(Remarks to the Author)

Reviewer #5

(Remarks to the Author)

Version 1:

Reviewer comments:

Reviewer #1

(Remarks to the Author)

I have reviewed the revised manuscript, and the updated version addresses most of the concerns raised, which makes this manuscript OK for publication.

Reviewer #2

(Remarks to the Author)

The Authors have successfully addressed all the concerns and suggestions raised in my review. Their revisions are clear and thoughtful. Consequently, I recommend the manuscript for publication in its current form.

Reviewer #3

(Remarks to the Author)

This revised manuscript represents a significant improvement over the original submission. We commend the authors for their thorough and thoughtful revisions in response to the reviewers' feedback. The inclusion of additional experiments and more detailed methodological descriptions is particularly appreciated.

We have one remaining comment regarding opposing patterns of A27h/M neuron activity during forward/backwards crawling. The present study claims to have “discover[ed] a previously unknown role of M neurons.” While we agree that a circuit model is outside the scope of the present study, helping the reader to place the authors' findings in the context of the current circuit literature seems appropriate. If the authors would add a version of their response regarding possible circuit explanations, including a reference to Jonaitis et al, to the paper's discussion, we believe that would be sufficient.

One smaller point: axis and legend labels appear to be missing from the new figure 5.

Reviewer #4

(Remarks to the Author)

Reviewer #5

(Remarks to the Author)

Responses to Reviewers' Comments for Manuscript NCOMMS-24-42413-T

CRASH2p: Closed-loop Two Photon Imaging in a Freely Moving Animal

Addressed Comments for Publication to

by

Paul McNulty, Rui Wu, Akihiro Yamaguchi, Ellie S. Heckscher, Andrew Haas, Amajindi Nwankpa, Mirna Mihovilovic Skanata, and Marc Gershow

Please find enclosed the revised version of our previous submission now entitled “CRASH2p: Closed-loop Two Photon Imaging in a Freely Moving Animal” with manuscript number NCOMMS-24-42413-T.

We would like to thank the reviewers for the careful reading and valuable comments on our manuscript. We are pleased to submit this revised version responding to these comments and suggestions. Please note the new figure showing recording from a central brain cell body, as requested by Reviewer 3. A summary of main modifications and a detailed point-by-point response to the comments from Reviewers 1 to 3 (following the reviewers’ order in the decision letter) follow.

Note: To enhance legibility, original comments are typeset in boxes. We apologize that hyperlinks to the main text will not function, but hyperlinks within this letter should. In our responses, we highlight additions or changes to the manuscript, and we separately provide a redlined version showing all changes to the text. Due to limitations of the Overleaf software package, there may be minor inconsistencies between these versions for which we apologize; in these cases, the final submitted manuscript is definitive.

Authors' Response to Reviewer 1

General Comments. Based on existing two-photon tracking methodologies, this study enhances capabilities with dual-color imaging, novel scanning patterns, and employs a new registration algorithm. These advancements enable three-dimensional two-photon imaging of freely moving *Drosophila* larvae in agarose, revealing neural patterns associated with behavior.

Response: Thank you for your careful reading of our manuscript

Please see our detailed responses below.

Comment 1

The title "Closed-loop Two Photon Imaging in Freely Moving Animals" inaccurately suggests that experiments were conducted on a variety of animals, while the study only experimented on *Drosophila* larvae. It would be clearer to state "Freely Moving *Drosophila* larvae in agarose" to specify the experimental conditions and eliminate ambiguity.

Response: Thank you for the comment.

We apologize that 'animals' was ambiguous as to whether multiple species of animal or multiple animals of the same species were studied in this work. In response to your concern, and that of Reviewer 2, we have amended the title to eliminate this ambiguity: 'CRASH2p: Closed-loop Two Photon Imaging in a Freely Moving Animal.'

We also note that the larvae were not 'in agarose' but were freely crawling *on* an agarose gel substrate, the standard substrate for behavioral experiments in larvae. This is a technical advance over prior methods in which animals crawl on glass or are sandwiched between agar and glass. We added a section to the methods (8.1.1) describing preparation of the agarose coated coverslips. Thank you for drawing our attention to this oversight.

Comment 2

In introduction, it is mentioned that “there are a variety of methods in freely moving transparent organisms, including zebrafish and *C. elegans*. But these techniques have not transferred to freely behaving larval *Drosophila*, and this model is more difficult to image during freely behavior.” Please explain this why larval *Drosophila* is more difficult to image compared to zebrafish and *C. elegans*, as the two models are also moving fast. Also, does that mean their systems are deficient for imaging moving larval *Drosophila* in agarose.?

Response: Thank you for the question

In the supplemental discussion we write ‘But these methods have not translated to larval *Drosophila*, likely due to the complicated motion of the larva which translates and distorts the brain, scattering and autofluorescence of the cuticle whose optical properties change over the course of a crawling stride, and the proximity to the brain of autofluorescent and scattering structures like the gut and trachea.’

To amplify these points: The fact that larvae are imaged in air rather than in water exacerbates scattering of the cuticle due to index mismatch. The larva’s peristaltic crawling results in segment-by-segment compression of the cuticle, which means that the scattering changes in a time-dependent manner correlated with behavior. The larva’s gut goes through its brain. The brain is connected to the mouth hooks [1], which the larva uses as anchors during forward crawling [2]. As a result the brain moves internally, often opposite the body motion, and in 3-dimensions; it is also compressed and bent by external motion [3].

Many of the structures proximate to the larva’s brain, including the mouth hooks, gut, trachea, and cuticle are auto-fluorescent to some degree. For single-photon imaging techniques, this presents a confounding background that fluctuates in intensity depending on the movement of the autofluorescent structures. The signal due to background autofluorescence cannot be removed by ratiometric correction.

While we cannot say without using them that none of the systems deployed for fish or for worms could be adapted to record from single neurons within the larva’s CNS, we can say that there is no published work to support the idea that they can.

Comment 3

Lines 81-83 state that the authors' previous work introduced a two-photon tracking microscope to record activity from individual neurons in unrestrained and freely crawling *Drosophila* larvae. In lines 84-86, it is mentioned that two-photon recording in unrestrained animals remained elusive for recording populations of neurons because two-photon microscopy is sensitive to motion. Please explain why recording populations of neurons is more challenging compared to imaging single neurons, and what has been done in this work to solve the difficulties.

Response: Thank you for the question

Our first tracking microscope [4] was capable of tracking one or two (closely separated) cell bodies for recording, but was limited by the physical inertia of the scanning galvanometric mirrors. More recently, using acousto-optic deflectors (AODs), we were able to record from several widely spaced neurons [5]. We expected to be able to follow many cell bodies with this microscope but found that we could not maintain tracking of many cells in crawling larvae. There is extensive discussion of the limitations of the microscope in this second publication. Essentially, the motion of the larva is too complicated for the simple (Kalman filter) algorithm we use to account correctly for the possible correlations between the various neurons' movements.

While these tracking microscopes were able to rapidly track and record activity from single cell bodies, neither was capable of forming images or of recording from processes. CRASH2p uses separate imaging and tracking scan heads to allow the recording of large (CNS-scale) two-color volumetric (3D) images with high temporal resolution, corrected in real-time for 3D translation, in behaving larvae. Additionally, in this work we demonstrate a new image registration and intensity correction pipeline developed to correct for rigid rotation and non-rigid deformations of the brain and fluctuations in the efficiency of exciting and recovering fluorescence. Together these allowed for the two photon recording of activity from populations of neurons with cellular resolution in freely behaving animals.

Please also see the additions to the manuscript in our response to your Comment 6

Comment 4

Line 116, the term “tracker” was used for the first time without any explanation or definition. Please clarify that.

Response: Thank you for the comment.

We revised the text from ‘We use one scan path of a dual-head microscope to track a reference neuron and the other to image the surrounding volume, using a scanning pattern designed to combat motion artifacts. Feedback from the tracker stabilizes the imaging path ...’ to

One scan path of a dual-head microscope operates a tracker to follow a reference neuron, while the other images the surrounding volume, using a scanning pattern designed to combat motion artifacts. Feedback from the tracker stabilizes the imaging path ...

Comment 5

Line 179-182, it is mentioned the tracked neuron was added to the control signal of the galvos to keep the imaged volume centered on the tracked neuron. Please provide a detailed explanation of how the XY translation stage and the control of Galvos work together.

Response: We apologize that this material was difficult to locate in the manuscript

In the FPGA Code section of the methods, we explain ‘During tracking, the FPGA added the estimated location of the tracked cell to the target imaging spot location; thus the imaged region was always centered on the tracked cell, compensating translational motion in real time.’

Section 8.6 Feedback explains, ‘Following each measurement, the estimated position of the measured neuron was updated. We used this updated location to calculate the location of the next measurement scan. Other layers of feedback served to keep the tracked neuron centered within the field of view. From fastest to slowest: The x-y galvo deflectors were directed to the estimated location of the tracked neurons (on the imaging

path, the estimated location of the tracked neuron was added as an offset to the x- and y-deflection of the scan galvos), the piezo positioner on the objective was set to bring the estimated z- location of the neuron to the natural focus of the objective, and the stage was moved in all three axes to bring the tracked neuron to the natural central position (galvos centered and objective piezo at half range) of the microscope.'

To answer the reviewer's question about 'XY' translation of the stage and control of the galvos: The tracking algorithm estimated the location of the tracked cell; the galvanometers on the tracking head executed a sampling circle about this estimated location, providing information that the tracking algorithm used to update the estimated location of the tracked cell; the stage moved to bring the tracked cell back to the center of the imaging system.

Meanwhile, the estimated location of the tracked cell was also added to the control signal for the galvanometers of the imaging head. This had the effect of making the origin of the imaging system always coincide with the location of the tracked cell, compensating for translation in real time.

Although the question only asks about 'XY,' please note that the tracking system and the stage compensated for 'Z' motion as well. The difference in axial feedback is that two components of the microscope generated the axial scan – one TAG lens on each scan head generated a 200 kHz resonant axial scan, while a piezo on the objective adjusted the center of the scan range at <100 Hz. The estimated z-location of the tracked cell was used to adjust the position of the piezo so that the tracked cell was always located at the natural focus of the objective. Because both tracking and imaging utilized the same objective lens, moving the piezo to compensate for the axial motion of the tracked cell compensated for the same axial motion on the imaging path. As with the XY, the stage was directed in Z to bring the tracked cell to the center of the imaging system.

We added clarifying text to section 2.1.1

To maintain both the tracked neuron and the imaged volume within the scanning range of the microscope, the position of a 3-axis stage was continuously adjusted to bring the tracked neuron to the center of the microscope's field of view.

and to section 8.6.

Stage position commands were generated by a PID control loop with a 25 ms update rate. Each axis was controlled separately; for all 3 axes the error signal was the displacement of the tracked neuron from the natural objective focus. PID parameters were tuned by adjusting the 100 micron step response of the system to respond quickly without overshooting. Note that the stage controller includes its own PID parameters to translate input control signals into motor controls; these were not adjusted.

Comment 6

There is a lack of detailed comparison between this work and the previous work. The novelty of this study lies not in introducing motion tracking for the first time but in innovatively improving upon previous motion tracking efforts by introducing dual-color imaging and a novel scan pattern. Therefore, it is crucial to clearly articulate these innovative improvements compared to previous efforts.

Response: Thank you for the comment

We have added a section (3.1) to the discussion clarifying the advantages of imaging over recording from individual cell bodies

CRASH2p uses our previously described tracking technique [4, 5] to provide real-time feedback that stabilizes image formation in behaving animals. In these earlier microscopes, we *recorded* both red and green fluorescence from the tracked cell body, using their ratio as a measure of the Ca⁺⁺ concentration in that cell body. Using AODs [5], we were able to record from several cell bodies simultaneously. The tracked neurons were continuously sampled at > kHz rates, providing a high-bandwidth and high fidelity read-out of these neurons' activities. However, these microscopes were unable to form *images* while tracking. CRASH2p was able to form two color volumetric images extending hundreds of microns, sampled at > 10 Hz and with cellular resolution, all while the larva crawled tens of mm at mm/s top speeds. Except for the experiments of Figure 5, the experiments described in this work recording simultaneously from dozens of neurons and/or their processes were beyond the capabilities of those (or any previous) microscopes.

CRASH2p can be used to record from neurons throughout the larval CNS, including in the brain; our recordings in the VNC most fully demonstrate the advantages of volumetric imaging over tracking alone (Single cell tracking was sufficient to extract the correlation between MDN activity and backwards crawling [5]). Activity in different VNC segments corresponds with different body segments, so a full understanding of how posture is encoded or controlled requires simultaneous recording from many genetically identical neurons across the VNC. In our work with the R36G02 line Figure 2,3, recording from processes allowed us to separate activity of A27h from that of M/A19f, despite their cell bodies touching.

Comment 7

Any criterion for choosing the target neuron?

Response:

The tracked cell body is optimally located in the center of the region to be imaged and physically separated from other cell bodies and other fluorescent structures.

Comment 8

In the caption of Figure 1, there is a statement: "Note that although the tracker output was not used in the assembly of the left images (b,c), the tracker was still required for their acquisition; otherwise the labeled cells would have moved outside the imaged volume during scanning." Please clarify how images were obtained without tracker correction. If the tracker was still required for their acquisition to prevent *Drosophila* larvae from moving outside the imaged volume, how were images obtained without tracker correction? Were the motion corrections obtained from the tracker applied to the collected images?

Response:

Briefly, the tracker was used during recording to allow the microscope to sample the volume containing the VNC (otherwise, the recording would have been of a portion of

the larva's body or empty space where the VNC previously was). The motion corrections obtained from the tracker **were not** applied to form the images in Figure 1(b,c) but were applied to form the images in (d,e).

We calculate the rate of photon emission using the time history of the location of the imaging galvos and the times (and tag lens phase) at which photons were emitted. The rate of emission for a voxel is the number of photons detected while the laser spot was in that voxel divided by the amount of time the laser spot resided in that voxel.

Following an experiment, the data stored on disk contains the FPGA clock time at which every photon was detected, the FPGA clock time of the synchronizing signals from the two tag lenses, the read-back locations of the galvos and piezo positioner at defined FPGA clock times, and other information used to synchronize the timing of stage readouts and IR ("behavior") camera frame. The process of converting these raw signals to volumetric images is described in section 8.7.

The raw data stored on disk contains the read-back locations of the imaging galvos, without regard to the tracker correction. One of the first steps in image reconstruction is to subtract the tracked neuron location from the imaging galvo positions to move the image into the reference frame of the neuron. To generate the image projections in Figure 1 b,c, we did not do this subtraction.

We now explain this directly in the methods section 8.7.2 'Determining photon emission location'

To generate the image projections without tracker correction shown in Figure 1b,c we used the raw locations without subtracting off the tracker location:

$$x(t) = x_{galvo}(t) \tag{1}$$

$$y(t) = y_{galvo}(t) \text{ Figure 1b,c only} \tag{2}$$

$$z(t) = A \cos(\phi(t)) \tag{3}$$

Comment 9

Using the red channel for correcting rotations, deformations, and intensity variations, and applying these corrections to the green channel, is advantageous. However, it is essential to specify the specific benefits of this approach compared to using only the green channel. Comparative experiments should demonstrate the effectiveness of using a red-color channel for correction and application to the green channel versus using the green channel alone. This comparison will illustrate the necessity of a dual-color imaging system. Given that dual-color systems increase system complexity, demonstrating the advantages of such corrections is crucial for justifying the construction of a dual-color imaging system.

Response: Thank you for the suggestion

Two-color imaging is *essential* for simultaneous tracking and recording of neural activity. Most popular Ca^{++} indicators, including the GCaMP series, are quite dim, especially when not binding Calcium. Tracking a neuron requires that the neuron be labeled with a bright, stable (i.e. not Ca^{++} sensitive) fluorophore. To allow simultaneous tracking and activity monitoring, this indicator must have different spectral properties than the Ca^{++} fluorophore. The ability to excite and record from two distinct colored indicators is essential to tracking and recording even a single neuron.

Our earlier work [4] showed that in many cases, the recovered fluorescence from a tracked neuron is strongly affected by movement associated with normal behavior (See e.g. figure S4a and video S4 in that work), but that ratiometric measurement could be used to recover signatures of activity independent of these effects.

We initially designed the CRASH2p microscope with the hope that the red fluorescence of the tracked neuron could be used to correct motion artifacts across an entire imaged volume. In fact, we found that motion affected different parts of the imaged volume differently (e.g. when the larva's CNS rotated around an axis orthogonal to the objective's, causing the direction and magnitude of the axial displacement to vary across the imaged volume) and that non-rigid deformations of the tissue further complicated imaging. Correcting these effects required that we form a complete reference image.

Beyond intensity correction, the red channel was required for registration. Again, the GCaMP indicators are dim in the off-state. While there would possibly be enough signal from inactive neurons for rigid registration using the GCaMP channel only, it would be

impossible to locate processes and cell bodies that had been deformed and/or translated due to non-rigid effects.

For an idea of the difficulties that would be created by using only the green channel without a red reference, we would invite the reviewer to consider movies M2 and M5, which show the green and red channels separately without correction. While in M2 the posterior-anterior wave is visible during forward crawling, a number of other intensity fluctuations are also evident, and the spacing between the cell bodies oscillates. During backward crawling, the ‘ladder’ can be seen in the red channel but it is completely absent in the green channel, so it would be impossible to locate the VOIs presented in Figure 2. Meanwhile in M5, there is significant side-to-side fluctuation in the magnitude of the green GFP signal that in the absence of the same fluctuations in the red channel, would have been interpreted as activity.

Although it might seem that dual-color imaging greatly increases the cost and complexity of the microscope, given the absolute need to excite and collect two fluorescent colors to allow simultaneous **tracking** (of a stable red indicator) and **imaging** (of a calcium sensitive green indicator), the extension to simultaneous tracking and **two-color imaging** requires only modest additional optics and electronics, including a low-cost FPGA-based demultiplexer we will detail more thoroughly in a separate publication.

We appreciate that this is quite a long response to an apparently simple request, but we hope that it will help the reviewer appreciate that it would not be straightforward to present ‘Comparative experiments’ that utilize only the green channel. ‘Comparative experiments’ could mean

1. Experiments in which only green fluorescence is excited and measured for both tracking and imaging – these are impossible, as discussed above. They would also require temporal multiplexing of the green channel, which is one of the difficulties the reviewer presumably seeks to avoid by single color imaging.
2. Experiments in which red fluorescence is used for tracking with separate green imaging – as we describe above, we began our investigations with this approach and the data proved exceedingly difficult to analyze, due to the need to perform non-rigid registration on dim (off) neurons. Further, it is clear from observation that there are significant fluctuations in intensity associated with motion, even in the control GFP experiments, which would contaminate these measurements.

3. ‘Experiments’ in which we reanalyze the existing data but pretend at some point not to have the red channel data. If we did not use the red imaging channel data at all, we would be unable to perform non-rigid registration and corresponding VOI analysis on the GCaMP samples. While it would be possible to use the red channel data for registration but not intensity correction, it is hard to see how this would address the reviewer’s concern about the ‘necessity of a dual-color imaging system,’ since the analysis still relies on dual-color images for registration. As noted previously, it is also clear from observation that for at least some experiments, there would be significant artifacts associated with motion that have been corrected by the red channel.

Of all the possible ‘comparative experiments,’ only a reanalysis which uses the two-color data for registration and VOI identification but then does not use the red channel for intensity correction is feasible. But this would in no way address the issue of the necessity of two-color imaging. Further, as seen in Comment 8, it can generate confusion when we perform an analysis that uses some but not all available information. For these reasons, we are regrettably unable to perform the requested experiments.

Comment 10

Minor questions: 1. Please add axis label in Figure 1(b)-(e) for clarity. 2. Line 596-599, it is mentioned “ these results show that we recorded distributed activity with sub-cellular resolution.....”. Please explain why “sub-cellular resolution” is achieved. 3. The quality of the figures needs to be improved.

Response: Thank you for the comment.

1. We added axes labels to Fig 1(e), corresponding to new axes labels also added to Fig 1(a).
2. We record from separately from different compartments of the same cell (cell bodies and processes).
3. We apologize that the submission system imposes a maximum file size that results in reduced quality. For the final publication we will submit the original line-art.

Authors' Response to Reviewer 2

General Comments. In this manuscript the Authors present the development of a closed-loop resonant axial-scanning high-speed two photon microscope able to acquire nervous activity in real-time and in 3D in unrestrained *Drosophila* larvae. The Authors used this instrument to study in great detail the nervous activity in the ventral nerve cord during strides. The Authors convincingly illustrated the advantages brought by their new technique in the feasibility of this kind of measure.

The scientific content is valid and original and it is of interest by the biological and the imaging community. The technique developed is ingenious and demonstrates high technical capabilities.

Therefore, I suggest the Editor to consider this manuscript for publication.

Response: Thank you for your positive assessment of our work

Please see our responses to your specific comments below.

Comment 1

It is cumbersome to find in the text information about the field-of-view dimensions (specified in the section 8.8.1) and volumetric imaging frequency. About the last point, the Authors state that pong scan allows to select the tradeoff between spatial resolution and temporal resolution offline, after the experiment. This is an interesting point. Nevertheless, it is cumbersome to find in the text the actual (volumetric) temporal resolution which was used in the final analysis. My advice is to put these information more in evidence in the text, making them more accessible to the reader.

Response: Thank you for the comment.

We added a section 8.3 Imaging Settings to the methods, following the expanded pong scan section (Comment 8), which also relates to tradeoffs between spatial and temporal sampling rates. This section includes a new table 1 that lists the sampling parameters for each figure in the paper (excluding the 20 experiments summarized in Figure 3b).

To make sure that information is most clearly available, at the end of the imaging settings section, we write

For registration and display purposes, volumes and their projections were reconstructed using a $1 \times 1 \times 2 \mu\text{m}$ x,y,z voxel and discretized into time bins of $1/2$ the characteristic raster time (see also section 8.7.3); for the main display figures and movies, this results in a time step of 94 ms. Temporal signals (e.g. Figure 2 b-e) were lowpassed by convolving with a Gaussian filter of standard deviation of 1 time step (= $1/2$ characteristic raster time).

And we expanded section 8.7.3 to also include this information

For movies and other 4D (time + volume) data sets, we constructed 4D histograms of counts and dwell times (following the motion correction steps described below, unless otherwise stated). Each 4D voxel was $1 \mu\text{m} \times 1 \mu\text{m} \times 2 \mu\text{m} \times (1/2\tau_{\text{raster}}$, or 94 ms) in x,y,z , and t , respectively. Unsampled locations had both 0 counts and 0 dwell time. These histograms were convolved with a 4D separable Gaussian filter with a standard deviation of 1 voxel size in each dimension; the rate was found by dividing the convolved count histogram by the convolved dwell time.

Comment 2

The manuscript makes a convincing argument for the validity of the presented method for the imaging of *Drosophila* larva. However, no other animal was tested. Therefore, the reader can't assess the generalizability of this method to image the nervous activity for different animal models. The Authors do not discuss this point adequately. This issue is exacerbated by the manuscript title, which states: "imaging in freely moving animals", suggesting a generalization which is not really present in the text.

Response: Please see our response to Reviewer 1's Comment 1

Comment 3

Time-space projection of ratiometric activity measures are present in several figures of the manuscript, e.g. Fig 2c. These images represent a “carpet” generated by mapping an 1-D average projection of space on the vertical axis and the time on the horizontal axis. This approach was used by the Authors to avoid potential experimenter bias introduced by ROIs drawing. This is actually a good idea. However, these images were not analyzed in a quantitative manner, instead they are only discussed in a qualitative way. This frustrates the advantage of a bias-resistant approach, therefore the Authors should analyze them in a quantitative way. Several different approaches are possible. My suggestion is to perform a 2D power spectrum (i.e. the modulus of the 2D Fourier transform squared) of the “carpet”. From the 2D power spectrum it is then possible to retrieve the average inclination and intensity of the “tilted bright lines” for the different conditions. Since each stride is of a different duration, this approach would probably work better if the time on the horizontal axis would be replaced by the phase “clock”.

Response: Thank you for the suggestion

The suggested analysis is very similar to the PCA analysis already presented. In the PCA analysis, we

1. Resample the data according to the phase clock, mapping time onto a $[0, 2\pi)$ interval.
2. Project the green and red fluorescence onto three axes ($x, y,$ and ϕ).
3. Using PCA, find a complex quasi-sinusoidal temporal signal corresponding to the most dominant frequency in the dataset.
4. Plot the amplitude and phase of this component, distributed over x and y .

The proposed analysis would

1. Resample the data according to the phase clock, mapping time onto a $[0, 2\pi)$ interval.
2. Project the green and red fluorescence onto two axes (box length and ϕ).
3. Calculate the 2D Fourier transform as a function of wave number k and frequency ω

4. Analyze the magnitude of the resulting transform

In fact, the 2D power spectrum is more difficult to interpret than the simple average projection.

Consider the general form of a traveling wave or pulse - $g(x, t) = f(x - vt)$ and calculate the Fourier transform $G(k, \omega)$

$$G(k, \omega) = \frac{1}{2\pi} \int_{-\infty}^{\infty} \int_{-\infty}^{\infty} e^{-ikx} e^{-i\omega t} f(x - vt) dx dt \quad (4)$$

$$= \frac{1}{2\pi} \int_{-\infty}^{\infty} dt e^{-i\omega t} \int_{-\infty}^{\infty} dx e^{-ik(x+vt)} f(x) \quad (5)$$

$$= \frac{1}{2\pi} \int_{-\infty}^{\infty} dt e^{-i(\omega+kv)t} \int_{-\infty}^{\infty} dx e^{-ikx} f(x) \quad (6)$$

$$= \sqrt{2\pi} \delta(\omega + kv) F(k) \quad (7)$$

where $F(k)$ is the Fourier transform of $f(x)$.

So the 2D Fourier transform of a traveling pulse is a 1D Fourier transform projected onto a tilted line. This line would not be easier to analyze than the original time-space projection, which also contains a tilted line.

In fact, our signal has additional spatial structure due to the ‘ladder rungs’ of the processes and would be represented by the product of a traveling signal and a time-invariant spatial structure function

$$h(x, t) = g(x, t) * s(x) = f(x - vt) * s(x) \quad (8)$$

The Fourier transform $H(k, \omega)$ would be the convolution of the two Fourier transforms

$$H(k, \omega) = G(k, \omega) \circ S(k) \delta(\omega) \quad (9)$$

which would appear as a series of tilted lines replicated across the k -axis depending on the Fourier transform of the distribution of the processes.

Comment 4

The usual DF/F0 transform for the fluorescence traces does not seem to be used in the manuscript. The Author should comment about this choice.

Response:

In all cases, unless stated otherwise, we apply a correction to both the red and green fluorescence rates to minimize fluctuations in the red channel, subject to spatial smoothness constraints on the correction.

For VOI analysis we used R/R_0 (R is the instantaneous ratio of green to red fluorescence, while R_0 is a baseline value of the ratio). Without fluctuations in the red channel, this is the same as a F/F_0 measure of green (GCaMP) fluorescence. Because the red indicator is not calcium sensitive, fluctuations in red intensity likely reflect motion-associated changes in the optical path between the objective and focal spot that also affect the recovered green fluorescence. Dividing the green intensity by the red intensity normalizes out these motion-associated fluctuations and provides a better measure of Ca^{++} associated changes in the fluorescence efficiency of the GCaMP indicator.

For images and time-space projections, a voxel-by-voxel baseline division is inappropriate because many voxels do not contain fluorescent indicator and have a baseline due to noise only. For these images we either present the rates themselves or divide by a single baseline (the median value for the data set shown) for the entire image.

The difference between R/R_0 and $\Delta R/R_0$ is simply whether the origin is at 0 or 1.

Comment 5

Regarding section 8.8.5 “PCA analysis of travelling waves”, I identify several issues:

a. “If a traveling wave is the dominant feature in a data set, then the first two principal components will be sinusoids 90 degrees out of phase;” Can the Authors give a justification for this statement? A mathematical tractation or, at least, a reference to another article explaining this relation.

b. Regarding the right side of equation 16: does the “R” symbol means the “real part”? Shouldn’t it be instead the modulus of the complex number? Why is the point value z present as a complex conjugate? Isn’t it a real number?

c. The last paragraph of this section starts at line 1513 (“To quantify the presence of an activity wave...”). It is not clear to me which part of the “Result” section is referred to by this part of the “Methods” section, maybe the last part of section 2.2.4 (from line 566)? If that is so, then I am not convinced by this method. The fraction of explained variance in the two conditions is compared starting from two models obtained from the independent best fit of two variables. Therefore, the fraction of explained variance now depends both from the starting condition of the two different conditions and from the results of the fitting procedure. Was the second factor adequately controlled by the Authors? It could introduce additional variability in an analysis that is presented as quantitative. Moreover, this procedure generates mathematically-strange results such as a negative variance: a quantity that is defined as not-negative by definition. Therefore, it is difficult for me to trust such an analysis method without a proper and extensive validation. Nevertheless, is this comparison really necessary? It seems to me an overcomplicated approach to try to quantify a phenomenon that is already qualitatively evident from the generated images (GCaMP vs GFP).

Response:

a. A traveling wave $f(\vec{x}, t)$ with spatially dependent amplitude can be written in the form

$$f(\vec{x}, t) = A(\vec{x}) \cos(\vec{k} \cdot \vec{x} - \omega t + \phi) \quad (10)$$

$$= A(\vec{x}) \cos(\vec{k} \cdot \vec{x}) \cos(\omega t + \phi) + A(\vec{x}) \sin(\vec{k} \cdot \vec{x}) \sin(\omega t) + \phi \quad (11)$$

$$= a_1(\vec{x}) \cos(\omega t + \phi) + a_2(\vec{x}) \sin(\omega t + \phi) \quad (12)$$

So the signal at any point in space is the weighted sum of two sinusoidal signals 90 degrees out of phase. ϕ can be split arbitrarily between the temporal ($\cos(\omega t + \phi)$) and

spatial $(\cos(\vec{k} \cdot \vec{x}) + \phi)$ terms, so any two sinusoids 90 degrees out of phase can be used; the choice of particular sinusoids fixes the temporal origin.

If $F_{ij} \equiv f(\vec{x}_i, t_j)$, then

$$F = AC \quad (13)$$

$$(A_{i1}, A_{i2}) = (a_1(\vec{x}_i), a_2(\vec{x}_i)) \quad (14)$$

$$\begin{pmatrix} C_{1j} \\ C_{2j} \end{pmatrix} = \begin{pmatrix} \cos(\omega t_j + \phi) \\ \sin(\omega t_j + \phi) \end{pmatrix} \quad (15)$$

Now assume that the signal is corrupted by noise or other signal E , with temporal mean 0, and offset by a temporally constant, spatially varying baseline \bar{F} ,

$$F = AC + E + \bar{F} \quad (16)$$

PCA on this matrix will generate a list of orthogonal vectors that form a basis for $F - \bar{F}$, ordered so that the square distance between $F - \bar{F}$ and a projection of $F - \bar{F}$ onto the first k vectors is the minimum of any set of k vectors.

Assuming the sampling times t_j are distributed evenly over at least one wave period, the two rows (c_1 and c_2) of C will be orthogonal to each other. The two rows of C completely explain the variance of AC . If the signal is ‘dominated’ by a traveling wave, we mean that most of the variance in the signal is due to the traveling wave component. In this case, the two rows of C are two orthogonal vectors that explain most of the variance of F and will be revealed by PCA.

While this technique is widely used, a derivation is found in this phase shifting interferometry paper [6].

b. Yes, the real part is intentional. Begin with a traveling wave of spatially varying amplitude,

$$f(\vec{x}, t) = A([\vec{x}]) \cos(\vec{k} \cdot \vec{x} - \omega t + \phi) \quad (17)$$

$$= a_1 \cos(\omega t + \phi) + a_2 \sin(\omega t + \phi) \quad (18)$$

$$= \Re\left((a_1(\vec{x}) + ia_2(\vec{x})) * e^{-i\omega t + \phi}\right) \quad (19)$$

$$= \Re(z(\vec{x})C(\Phi)) \quad (20)$$

$$\text{with} \quad (21)$$

$$z(\vec{x}) \equiv a_1(\vec{x}) + ia_2(\vec{x}) \quad (22)$$

$$C(\Phi) \equiv \cos(\Phi) + i \sin(\Phi), \quad \Phi \equiv \phi - \omega t \quad (23)$$

In Equation 28 of the main paper, $z(x, y)$ represents the projection onto the first two principal components ($a_1(x, y), a_2(x, y)$) as single complex number $z = a_1 + ia_2$. $|z(x, y)|$ represents the magnitude of the signal contained in the first two components (i.e. potentially associated with a traveling wave), while $\arg(z(x, y))$ represents the phase up to an additive constant.

c. We are sorry this was unclear. Our goal was to quantify how well the data from a particular experiment could be said to describe a traveling wave. While we agree that the presence/absence of a wave is qualitatively obvious from the image data (especially the movies), we also wanted to make a quantitative assessment, in line with the reviewer’s previous comment Comment 3. Further, we wanted to be able to summarize the results of multiple experiments. Figure 2g shows a comparison of 20 experiments on 20 different animals, which would be difficult to present via qualitative analysis of the images.

We chose a metric that poses the following question: ‘How much variance is explained if the PCA projection is forced to model a traveling wave?’ By construction, any PCA projection will explain as much variance as possible given the number of components allowed. So merely asking, ‘how much variance is explained by the first two principal components?’ would not be diagnostic of a traveling wave. Instead we first calculated the projection onto the first two principal components and represented them as a complex number

$$z(x, y) = a_1(x, y) + ia_2(x, y) = |z(x, y)|e^{i\phi(x, y)} \quad (24)$$

Then, without changing $|z(x, y)|$, we required that $\phi(x, y)$ have the form of a traveling wave

$$\phi(x, y) = k_x x + k_y y + \phi_0 \quad (25)$$

Given the original mean-subtracted signal $F - \bar{F}$ and the fixed values of $|z(x, y)|$ we found the values of k_x, k_y , and ϕ_0 that minimized the residual variance

$$\sum_{x, y, t} \left(f(x, y, t) - \bar{f} - |z(x, y)| \cos(k_x x + k_y y + \phi_0) c_1(t) - |z(x, y)| \sin(k_x x + k_y y + \phi_0) c_2(t) \right)^2 \quad (26)$$

where $c_1(t)$ and $c_2(t)$ are the first two principal components.

If the PC projection represented a traveling wave, then this adjusted projection would do fairly well at representing the data and would explain a substantial fraction of the variance. However, if the PC projection did not contain a traveling wave, the adjusted

projection would not explain the variance. In fact, the variance could even increase after subtracting off the best fit (Equation 26 above) because $|z(x, y)|$ were fixed to their original projected values.

Comment 6

Movies M4 and M5 are not explicitly referenced in the main text.

Response:

We now cite these movies when discussing the control experiment

Unlike for the larva with GCaMP labeled neurons, in GFP labeled larvae, we found minimal modulation of the green/red ratio with crawling; neither did we observe a traveling wave-like modulation of intensity associated with forward or backward crawling (Figure S3, Figure M4, Figure M5).

Authors' Response to Reviewer 3

General Comments.

In this paper, McNulty, et al. present a new microscopy technique, CRASH2p, which enables two photon imaging in moving animals. They demonstrate that their approach allows for the recording of population activity in the ventral nerve cord of *Drosophila* larvae with single neuron resolution. This is an important technique for systems neuroscience because *Drosophila* lead the field in many tools critical for dissecting neural circuits (e.g. single-cell targeting genetic tools, a whole-brain connectome, etc.) yet one important tool lacking in this animal is the ability for researchers to monitor cell-resolved neural activity in unrestrained behaving animals, as is possible in larger animals, for example, the mouse. This paper makes a significant advance towards addressing this significant need. The paper is well written, and their approach has the potential to open new avenues of investigation for matching behavior with its underlying neural origins using larval *Drosophila*. As such, this paper should be of significant interest to the general readership of Nature Communications.

Response: Thank you for your positive assessment of our work

Please see our responses to your specific comments below.

Comment 1

While impressive, CRASH2p does have limitations. The authors address some of the technical limitations in the discussion, but it would be helpful if they could discuss limitations on potential applications as well. One limitation is that the use cases of CRASH2p appear narrow, as imaging neural activity in small, freely moving, transparent organisms (eg. *C. Elegans*, zebrafish) does not require two photon microscopy, and larger model organisms (eg. mice) can support the weight of miniaturized microscopes. Secondly, CRASH2p does not allow for the imaging of larger or faster insects (for example, adult *Drosophila*), meaning that the use of this technology is limited to *Drosophila* larvae and other small, slow-moving organisms for which one-photon access to the entire nervous system is not possible.

Response:

Single-photon approaches have been much more successful in zebrafish larvae and *C. elegans* than in larval *Drosophila*, but two-photon microscopy is still of value in many cases, e.g. to avoid presenting a confounding visual stimulus, to reduce photobleaching, or to simplify the simultaneous use of optogenetic tools. See for instance discussion in [7] on the value of two photon imaging in *C. elegans*. Recent work on two photon recording in semi-immobilized zebrafish [8] shows the interest in two-photon recording in behaving fish as well.

While CRASH2p does not allow for recording from unrestrained walking flies (no technology currently allows recording from single cells in walking flies), the techniques demonstrated here could be used for motion correction in tethered flies, and also in other head-fixed or tethered preparations, including mice and fish. To date, the only two photon motion correction technique with performance similar to that of CRASH2p [8] uses a custom acousto-optic lens and requires implantation of reference beads; it is possible to imagine cases in which CRASH2p would be the preferred technique.

Comment 2

CRASH2p is introduced as a technology that allows for functional imaging in the larval CNS, but the proofs of concept provided by the authors are restricted to imaging in the VNC. Is imaging in the brain feasible with CRASH2p? Would it be possible for the authors to provide evidence of successful brain imaging with single neuron resolution?

Response: Thank you for the suggestion

We now present an additional set of experiments showing recording from MDN backwards-command neurons in the central brain, correlated with behavioral measurements. While there are two contacting MDN cell bodies on each side of the brain, they are usually treated as a single neuron for analysis. We chose MDN because the expected correlation of activity with behavior (higher activity during backward crawling) is straightforward to analyze and compare with prediction. We find that on an animal-by-animal basis and in aggregate, MDN is more active during backward than forward crawling, as expected.

We have previously reported this same result [5]. Generally, recording from individual cell bodies can be accomplished either with CRASH2p or our previous approaches, which is why the majority of our manuscript focuses on recording from distributed activity in cell bodies and processes throughout the VNC – these are the experiments that were previously impossible.

Comment 3

All the examples provided by the authors appear to be imaging from neuronal processes, and not cell bodies. Does CRASH2p have sufficient spatial resolution to identify and image activity in cell bodies? Can cell bodies be spatially resolved from one another?

Response:

In the R36G02 recordings, we visualize activity in A27h processes and cell bodies and in M (A19f) cell bodies. The A27h and M cell bodies touch each other and so provide a test of whether different activity can be recovered from neighboring cell bodies. In fact, when we characterize areas that are active during forward and backward crawling, we find non-overlapping spatial footprints. The most reasonable explanation for the data is that the backwards-active regions are M neuron cell bodies that are resolved separately from the A27h cells.

In general, recording from processes is more challenging than recording from cell bodies because of their small sizes and odd shapes. In addition to the ability to record from large volumes, one of the most exciting abilities of our microscope is the ability to record from processes in addition to cell bodies. These abilities also present key advances over our previous tracking microscopes, which could record from a few cell bodies only. For these reasons, we focused the manuscript on the novel ability to resolve distributed activity in processes, but recording from cell bodies is also possible.

Our new MDN recordings also demonstrate the ability to record from cell bodies. While two contacting cell bodies are necessarily treated as a single unit here (the position of the cell bodies with respect to each other is apparently not stereotyped -see confocal images in [9] - and it is unclear whether they would be expected to show different patterns of

activity during normal behavior), this shows that there is no obstacle to recording from cell bodies in the brain.

Comment 4

The single tracked neuron appears to be scanned continuously within a small ROI, raising concerns that photobleaching may degrade tracking quality over the course of an experiment. Could the authors provide data that quantifies the photobleaching of the tracker neuron over time? Further, could they quantify the relationship between bleaching and tracking quality to demonstrate that their tracking approach remains reliable? It would also be useful for readers to know at what point, if any, this becomes unreliable.

Response: Thank you for the question

In some experiments, we observed photobleaching of the tracked neuron consistent with our previous reports [4], while in others the neuron did not bleach, perhaps because less fluorescence was excited from the tracked neuron to begin with. We have added a section 8.5.1 to the methods quantifying this

In previous work, we observed a decrease in red fluorescence intensity in the tracked neuron due to photobleaching over time; typically after 15 minutes of tracking, the red fluorescence rate was reduced to 40% of its initial value [4]. An exponential fit to the red fluorescence rate of the tracked neuron, as measured by the tracker, for the experiment shown in Figure 2, found that the tracked neuron was reduced to 75% of its initial fluorescence over ~ 260 s of tracking. For the experiment in Figure 4, the final neuron's tracked fluorescence was reduced to 80% of the initial. Both of these results are consistent with previously measured rates of bleaching. The MDN experiments (Figure 5) showed no degradation in red fluorescence over 15 minutes of tracking, but also recovered lower raw rates of mCherry fluorescence from the tracked neuron. In contrast to our tracking-only microscopes ([4, 5]), where the red fluorescence of the tracked neuron was used for both spatial feedback and ratiometric correction of the green fluorescence signal, in CRASH2p, the tracked neuron's fluorescence is used only for spatial feedback during the actual experiment and not in later analysis.

Comment 5

The authors initially expect, given previous work, that A27h pre-motor interneurons will be silent during backward crawling (lines 309-311), but observe a front-to-back wave during backward crawling (Figure 2G, lines 412-414). This effect is much weaker than the back-to-front waves present during forward walking but is still present. They explain this by demonstrating different patterns of spatial activation during forward vs. backwards walking and suggest that the ROIs active during backwards walking are M neurons also labeled by their Gal4 line. While I find their PC analysis (Figure 3) compelling, their findings leave me puzzled as to how the broader circuit may be operating. Zeng et al 2021 (the authors' citation #65) found that A27h and M neurons are electrically coupled via gap junctions and show synchronous activity patterns. This seems incompatible with the authors' finding that A27h and M neurons are differentially recruited by opposing locomotor programs. Can the authors address this discrepancy?

Response:

While there are plausible explanations for why A27h is active during forward crawling and M active during backward crawling (the gap junctions present during development might not exist in the 2nd instar stage studied here; A27h/M neurons might be strongly inhibited during backwards/forwards locomotion respectively) building a circuit model to explain this phenomenon is beyond the scope of this paper, which focuses on the development and demonstration of a new microscopy technique.

While our work is the first to record from M/A19f neurons in behaving animals, the original A27h paper [10] used the same R36G02 line, and both A27h and M neurons are visible in Video 6 of that paper. In this recording, in an isolated CNS, M and A27h appear to have different activity patterns from each other during fictive forward and backward crawling. In a recent preprint, using a different driver line Jonaitis and co-workers [11] found that A19f (same as M) is active during both forward and backward fictive crawling; recall that Fushiki and coworkers found that A27h is inactive during fictive backward crawling. Thus our finding that M neuron is active during backward crawling is consistent with previous reports.

Jonaitis and co-workers [11] also found that A19f receives substantial input from cholinergic A18b, which in turn receives input from the Mooncrawler Descending Neuron, a

backwards command neuron. MDN inhibits A27h through GABAergic Pair1 neurons[9]. This provides a potential circuit explanation for why A19f/M is active during backward crawling but A27h is silent.

Note that although A19f as shown in [11] is apparently morphologically identical to M as shown in [12], the two papers use different driver lines to label the neurons. This paper uses R36G02, the same driver line used to label M in [12]. Our identification of A19f as M comes from the identification in [11], ‘Previous work has suggested that these neurons [A19f] could play a critical role in the formation of larval locomotor circuits early in development and this could reflect a role as motor activity monitors early in development[12]’

Comment 6

The authors’ identification of distinct cell types labelled by the R326G02-Gal4 line is based on the relative position of ROIs, which is not the most reliable method for cell type identification (especially when cell bodies and processes are highly overlapping, as appears to be the case for A27h and M neurons in citation #65). Can the authors unambiguously confirm the morphology or identity of recorded neurons (e.g. by comparing the targeted cell morphology to the connectome or other method)? From the connectome, are there any neurons with similar morphologies that the authors may be mistaking as A27h or M neurons? Because of the unexpected result for M neuron activity (see point #2 above), it is particularly important that the identification of these neurons is robust if the authors wish to draw any conclusion about M cell function during crawling.

Response:

As calcium imaging in living animals has a lower spatial resolution than techniques available only in fixed tissue (e.g. expansion microscopy, electron microscopy), linking calcium imaging experiments to subsequent fixed-tissue imaging of the same cells is an area of ongoing research but unfortunately beyond the abilities demonstrated in this paper.

The R36G02 line labels a limited number of neurons, and the morphology of A27h is distinct. Thus we are certain that the ladder like processes we analyze belong to A27h.

Given that Zeng and co-workers [12] identify "M" as a neuron also labeled by R36G02 with a cell-body directly touching A27h but with processes that do not cross the midline, we are certain that our recording contains both A27h and M neuron cell bodies.

It is clear from observation of activity during forward and backward crawling (movie M1) that there are forward (posterior-anterior) waves during forward crawling and backward (anterior-posterior) waves during backward crawling and that these waves occupy spatially distinct substrates. The effect is strong enough that it can be seen in the unregistered and uncorrected projections (movie M2). Analysis of the stride aligned data (movie M3, figure 3) strengthens this conclusion.

Thus we know the following

1. The midline-crossing processes that unambiguously belong to A27h show activity patterns expected of A27h - posterior-anterior waves during forward crawling and (near) silence during backward crawling
2. Cell bodies on the periphery that could belong to A27h or M show activity patterns not expected of A27h - anterior-posterior waves during backward crawling and suppressed activity during forward crawling.

Either A27h processes encode different (and in fact opposing) behaviors than A27h cell bodies, or M neurons are the backwards active neurons visualized in our experiments. We have previously found that A27h cell bodies are active during forward [4, 5], but not backward [4] crawling, consistent with measurements during fictive locomotion in isolated CNS [10].

Comment 7

Could the authors add discussion of how their findings relate to the function of the broader circuit during larval locomotion? Their results are interesting but likely difficult for the reader to contextualize with the provided information. Even with the main focus of the paper being the microscopy technology, a broader locomotion circuit discourse does seem lacking given the emphasis on this circuit and the asserted novelty of the findings.

Response: Thank you for the suggestion

We added a reference to review articles at the beginning of our discussion of imaging experiments in larvae (Section 2.2)

While the *Drosophila* larva is an excellent model for understanding the development and function of motor circuits [13, 14], the inability to visualize neural activity in behaving animals hinders progress,

Comment 8

The authors use a “pong” scan pattern which they claim improves image registration and reconstruction compared to a raster scan pattern or a Lissajous scan pattern. Given that this scan pattern appears to be a novel approach, can the authors provide quantification of these improvements?

Response: Thank you for the suggestion

We added a supplemental figure Figure S5 and expanded the discussion in the ‘Pong Scan’ section of the methods.

Pong scanning is unambiguously superior to Lissajous scanning at the same frequency, because the triangle wave more evenly distributes sampling over time, instead of concentrating the sampling on the periphery of the volume. Lissajous scanning can be used at higher frequencies, because resonant scanners, which allow > 8 kHz scan frequencies can only generate sinusoids. Given the need to sample complete axial lines using the resonant tag lens in CRASH2p microscopy, CRASH2p would not benefit from these higher frequencies, and therefore Pong scanning is superior in this application.

Figure S5 compares a pong scan to a raster scan for typical parameters used in our experiments: a 200 micron square with a target pixel size of 800 nm (the resonant TAG lens extends this to a 200x200x50 micron volume). A raster scan strategy completely samples this volume in $\tau_{raster} = 189$ ms, while a pong scan takes twice as long to completely sample at the same pixel density (Figure S5a). In 95 ms or $1/2\tau_{raster}$, both the pong scan and raster scan sample roughly half the 800 nm pixels, but the pong scan samples more uniformly across the image (Figure 2d),

and as a result visits every 2 micron pixel in the image at least once during the same time (Figure S5a, black dashed line). In 0.024s or $1/8\tau_{raster}$ (Figure S5c), the raster scan visits only 1 small band of the image, while the pong scan visits the entire image; this property allows for better correction of rotation (strictly local measurements cannot distinguish translation from rotation - see Figure 7a). Following a single raster scan, the portion of the volume first sampled by the raster would have grown ‘stale’ - it has not been sampled in over 100 ms. In contrast, following the same duration pong scan, most 4 micron regions of the plane (columns of the volume) would have been recently sampled (Figure S5f). This property of repeated sampling distributed more uniformly over space provides pong scanning an advantage in the reconstruction of time-varying fluorescence traces.

Note that when we discuss sampling a pixel, we mean that we have measured a location within the pixel, but not that we have measured the integrated fluorescence of the entire pixel. Sampling a 2 micron pixel means measuring at least one (PSF-sized) location within that pixel. So it would be possible to miss fine structures within a large pixel in any given sampling pass.

Note also that regardless of scan method, motion associated with behavior makes *a priori* targeting of sampling locations on micron scales difficult, although the locations can be reconstructed after the fact using our registration pipeline.

References

1. Wu, J. S. & Luo, L. A protocol for dissecting *Drosophila melanogaster* brains for live imaging or immunostaining. en. *Nature Protocols* **1**, 2110–2115. ISSN: 1754-2189, 1750-2799 (Nov. 2006).
2. Heckscher, E. S., Lockery, S. R. & Doe, C. Q. Characterization of *Drosophila* Larval Crawling at the Level of Organism, Segment, and Somatic Body Wall Musculature. en. *The Journal of Neuroscience* **32**, 12460–12471. ISSN: 0270-6474, 1529-2401 (Sept. 2012).
3. Sun, X. & Heckscher, E. S. Using Linear Agarose Channels to Study *Drosophila* Larval Crawling Behavior. *JoVE (Journal of Visualized Experiments)*, e54892–e54892. ISSN: 1940-087X (Nov. 2016).

4. Karagyozev, D., Mihovilovic Skanata, M., Lesar, A. & Gershow, M. Recording Neural Activity in Unrestrained Animals with Three-Dimensional Tracking Two-Photon Microscopy. *Cell reports* **25**, 1371–1383.e10. ISSN: 2211-1247 (Oct. 2018).
5. Yamaguchi, A. *et al.* Multi-neuronal recording in unrestrained animals with all acousto-optic random-access line-scanning two-photon microscopy. *Frontiers in Neuroscience* **17**. ISSN: 1662-453X (2023).
6. Vargas, J., Quiroga, J. A. & Belenguer, T. Analysis of the principal component algorithm in phase-shifting interferometry. EN. *Optics Letters* **36**. Publisher: Optica Publishing Group, 2215–2217. ISSN: 1539-4794 (June 2011).
7. Schrödel, T., Prevedel, R., Aumayr, K., Zimmer, M. & Vaziri, A. Brain-wide 3D imaging of neuronal activity in *Caenorhabditis elegans* with sculpted light. en. *Nature Methods* **10**, 1013–1020. ISSN: 1548-7091 (Oct. 2013).
8. Griffiths, V. A. *et al.* Real-time 3D movement correction for two-photon imaging in behaving animals. en. *Nature Methods* **17**, 741–748. ISSN: 1548-7091, 1548-7105 (July 2020).
9. Carreira-Rosario, A. *et al.* MDN brain descending neurons coordinately activate backward and inhibit forward locomotion. *eLife* **7** (eds Calabrese, R. L. & Marder, E.) e38554. ISSN: 2050-084X (Aug. 2018).
10. Fushiki, A. *et al.* A circuit mechanism for the propagation of waves of muscle contraction in *Drosophila*. en. *eLife* **5**, e13253. ISSN: 2050-084X (Feb. 2016).
11. Jonaitis, J. *et al.* *Steering From the Rear: Coordination of Central Pattern Generators Underlying Navigation by Ascending Interneurons* en. Pages: 2024.06.17.598162 Section: New Results. June 2024.
12. Zeng, X. *et al.* An electrically coupled pioneer circuit enables motor development via proprioceptive feedback in *Drosophila* embryos. en. *Current Biology* **31**, 5327–5340.e5. ISSN: 09609822 (Dec. 2021).
13. Hunter, I., Coulson, B., Zarin, A. A. & Baines, R. A. The *Drosophila* Larval Locomotor Circuit Provides a Model to Understand Neural Circuit Development and Function. English. *Frontiers in Neural Circuits* **15**. Publisher: Frontiers. ISSN: 1662-5110 (July 2021).
14. Kohsaka, H. Linking neural circuits to the mechanics of animal behavior in *Drosophila* larval locomotion. English. *Frontiers in Neural Circuits* **17**. Publisher: Frontiers. ISSN: 1662-5110 (Aug. 2023).

Based on existing two-photon tracking methodologies, this study enhances capabilities with dual-color imaging, novel scanning patterns, and employs a new registration algorithm. These advancements enable three-dimensional two-photon imaging of freely moving *Drosophila* larvae in agarose, revealing neural patterns associated with behavior. However, this manuscript faces some issues.

Major questions:

1. The title "Closed-loop Two Photon Imaging in Freely Moving Animals" inaccurately suggests that experiments were conducted on a variety of animals, while the study only experimented on *Drosophila* larvae. It would be clearer to state "Freely Moving *Drosophila* larvae in agarose" to specify the experimental conditions and eliminate ambiguity.

2. In introduction, it is mentioned that "there are a variety of methods in freely moving transparent organisms, including zebrafish and *C. elegans*. But these techniques have not transferred to freely behaving larval *drosophila*, and this model is more difficult to image during freely behavior." Please explain this why larval *drosophila* is more difficult to image compared to zebrafish and *C. elegans*, as the two models are also moving fast. Also, does that mean their systems are deficient for imaging moving larval *drosophila* in agarose.?

3. Lines 81-83 state that the authors' previous work introduced a two-photon tracking microscope to record activity from individual neurons in unrestrained and freely crawling *Drosophila* larvae. In lines 84-86, it is mentioned that two-photon recording in unrestrained animals remained elusive for recording populations of neurons because two-photon microscopy is sensitive to motion. Please explain why recording populations of neurons is more challenging compared to imaging single neurons, and what has been done in this work to solve the difficulties.

4. Line 116, the term "tracker" was used for the first time without any explanation or definition. Please clarify that.

5. Line 179-182, it is mentioned the tracked neuron was added to the control signal of the galvos to keep the imaged volume centered on the tracked neuron. Please provide a detailed explanation of how the XY translation stage and the control of Galvos work together.

6. There is a lack of detailed comparison between this work and the previous work. The novelty of this study lies not in introducing motion tracking for the first time but in innovatively improving upon previous motion tracking efforts by introducing dual-color imaging and a novel scan pattern. Therefore, it is crucial to clearly articulate these innovative improvements compared to previous efforts.

7. Any criterion for choosing the target neuron?

8. In the caption of Figure 1, there is a statement: "Note that although the tracker output was not used in the assembly of the left images (b,c), the tracker was still required for their acquisition; otherwise the labeled cells would have moved outside the imaged volume during scanning." Please clarify how images were obtained without tracker correction. If the tracker was still required for their acquisition to prevent *Drosophila* larvae from moving outside the imaged volume, how were images obtained without tracker correction? Were the motion corrections obtained from the tracker applied to the collected images?

9. Using the red channel for correcting rotations, deformations, and intensity variations, and applying these corrections to the green channel, is advantageous. However, it is essential to specify the specific benefits of this approach compared to using only the green channel. Comparative experiments should demonstrate the effectiveness of using a red-color channel for correction and application to the green channel versus using the green channel alone. This comparison will illustrate the necessity of a dual-color imaging system. Given that dual-color systems increase system complexity, demonstrating the advantages of such corrections is crucial for justifying the construction of a dual-color imaging system.

Minor questions:

1. Please add axis label in Figure 1(b)-(e) for clarity.
2. Line 596-599, it is mentioned " these results show that we recorded distributed activity with sub-cellular resolution.....". Please explain why "sub-cellular resolution" is achieved.
3. The quality of the figures needs to be improved.